# A short intrinsically disordered region at KtrB's N-terminus facilitates allosteric regulation of K$^+$ channel KtrAB

Janina Stautz[1,8], David Griwatz [1,8], Susann Kaltwasser [2], Ahmad Reza Mehdipour[3,4], Sophie Ketter[1], Celina Thiel[1], Dorith Wunnicke[1], Marina Schrecker[1], Deryck J. Mills [5,7], Gerhard Hummer [4,6], Janet Vonck [5,9] ✉ & Inga Hänelt [1,9] ✉

K$^+$ homeostasis is crucial for bacterial survival. The bacterial K+ channel KtrAB is regulated by the binding of ADP and ATP to the cytosolic RCK subunits KtrA. While the ligand-induced conformational changes in KtrA are well described, the transmission to the gating regions within KtrB is not understood. Here, we present a cryo-EM structure of the ADP-bound, inactive KtrAB complex from *Vibrio alginolyticus*, which resolves part of KtrB's N termini. They are short intrinsically disordered regions (IDRs) located at the interface of KtrA and KtrB. We reveal that these IDRs play a decisive role in ATP-mediated channel opening, while the closed ADP-bound state does not depend on the N-termini. We propose an allosteric mechanism, in which ATP-induced conformational changes within KtrA trigger an interaction of KtrB's N-terminal IDRs with the membrane, stabilizing the active and conductive state of KtrAB.

In prokaryotic cells, potassium ions (K$^+$) are the most abundant intracellular cations and fulfill a broad spectrum of tasks. They play an essential role in different cellular processes, including osmoregulation, pH homeostasis, regulation of protein synthesis, enzyme activation, membrane potential adjustment, and electrical signaling in biofilms[1–4]. Therefore, K$^+$ homeostasis is essential for cell survival, and K$^+$ uptake and release need to be regulated tightly. For the translocation of K$^+$ across the membrane, cells contain several types of transporters and channels, each possessing different molecular properties and conditions for activation and inactivation[5].

Many prokaryotes encode for one or more homologs of 2-TM potassium channels like KcsA or MthK, which form a tetrameric pore. Each protomer is built of two transmembrane (TM) helices (M1/M2) and a pore loop (P) in between them, also referred to as M1PM2 domain[6,7]. Over 50% of these channels are regulated by ligand binding

to cytosolic RCK (Regulator of Conductance of K$^+$) domains, which form a regulatory octameric gating ring[8–11]. Upon ligand binding in MthK or GsuK, for example, large conformational changes of the gating ring result in the movement of a flexible linker that covalently connects the pore domain and the RCK domain[12]. In turn, this induces conformational rearrangements in the pore region. Particularly, a helical bundle is pulled open, leading to a pathway for K$^+$ flux. To avoid excessive potassium fluxes, of which both influx and efflux would induce cytotoxic processes, many 2-TM K$^+$ channels are inactivated by N- and/or C-type inactivation only milliseconds after activation[13]. While functionally well characterized, the physiological role of most bacterial 2-TM channels remains elusive.

Physiologically better understood systems are the uncommon RCK-gated channels KtrAB and TrkAH, which substantially differ from classical 2-TM potassium channels. They are the major potassium

[1]Institute of Biochemistry, Goethe University Frankfurt, Frankfurt am Main, Germany. [2]Central Electron Microscopy Facility, Max Planck Institute of Biophysics, Frankfurt am Main, Germany. [3]Center for Molecular Modeling, Ghent University, Zwijnaarde, Belgium. [4]Department of Theoretical Biophysics, Max Planck Institute of Biophysics, Frankfurt am Main, Germany. [5]Department of Structural Biology, Max Planck Institute of Biophysics, Frankfurt am Main, Germany. [6]Institute for Biophysics, Goethe University Frankfurt, Frankfurt am Main 60438, Germany. [7]Deceased: Deryck J. Mills [8]These authors contributed equally: Janina Stautz, David Griwatz. [10]These authors jointly supervised this work: Janet Vonck, Inga Hänelt. ✉e-mail: janet.vonck@biophys.mpg.de; haenelt@biochem.uni-frankfurt.de

uptake systems of bacteria and prevent, for example, water loss and subsequent plasmolysis during hyperosmotic stress[14–16]. In contrast to 2-TM channels, KtrAB and TrkAH comprise two individual subunits, KtrB/TrkH and KtrA/TrkA, without a covalent tether[17–19]. The transmembrane subunits KtrB/TrkH form homodimers, with each protomer comprising four nonidentical, covalently fused M1PM2 domains (D1-D4), resulting in two individual pseudo-tetrameric pores for ion translocation. Compared to the canonical $K^+$ selectivity filter sequence TVGYG, only the first glycine residue in each pore loop is preserved in KtrB and TrkH, suggesting less selective ion translocation[20,21]. In agreement with this, single channel recordings of TrkAH revealed conductivity for all tested monovalent cations[19]. In contrast, for KtrAB it was shown that under native-like mixed-ion conditions, $K^+$ is selectively accumulated in a $Na^+$-dependent manner[22]. A gating helical bundle, as found in many 2-TM $K^+$ channels, is missing in KtrB and TrkH. Instead, an intramembrane loop just below the selectivity filter, formed by the central part of broken helix D3M2, and a highly conserved arginine from helix D4M2 are proposed to function as channel gate both in TrkAH and KtrAB[23–25].

The gating of the intramembrane loops is thought to be controlled by the cytosolic regulatory subunits KtrA/TrkA. They assemble into a ring of eight RCK domains in complex with the dimeric pore-forming subunits. For both systems, nucleotide-triggered conformational changes within the RCK ring were shown to regulate channel activity[17–19,26,27]. In TrkAH, where the TrkA ring is formed by a tetramer of fused RCK1-RCK2 domains, selective ATP binding to the four RCK2 domains breaks the ring apart into two dimers, pulling on the broken helices D3M2b such that the intramembrane loops are opened. The exchange of ADP to ATP in RCK1 supports pore opening by moving the pore-lining helices[27]. In the KtrAB system, the regulatory mechanism deviates, as the ring is a homooctamer of KtrA subunits and interacts differently with the KtrB dimer. In the ADP-bound conformation, the KtrA ring is oval-shaped, while in the presence of ATP it adopts a square-shaped conformation[28,29]. A low-resolution cryo-EM map of the KtrAB complex from *Vibrio alginolyticus* showed an extension of the D1M2 helices of KtrB into the oval-shaped KtrA ring, establishing direct contact between both subunits in the ADP-bound conformation[18]. The extended helices appeared to lock the intramembrane loops in a pore-blocking conformation. Based on this map and a first ATP-bound structure of KtrAB from *B. subtilis* (*Bs*KtrAB)[17], a gating mechanism was suggested: The ATP-bound, square-shaped KtrA ring displaces the

extended D1M2 helices, leading to their transition into helical hairpins[17,18,29]. In combination with conformational changes of the pore-lining helices and the ensuing reorientation of the conserved arginines from the pores, the intramembrane loops unlock, allowing $K^+$ flux. A recent study with structures of *Bs*KtrAB in the presence of ADP and ATP extended this model, showing that the binding of ATP is mediated by $Na^+$ rather than by $Mg^{2+}$ as previously suggested, explaining the $Na^+$ dependency of KtrAB activation[30]. However, despite the formation of helical hairpins by the extended D1M1 helices observed in both available ATP-bound structures, the intramembrane loops do not adopt an active, open state as in TrkAH but instead still block the pores[17,30]. Thus, notwithstanding the available structural data, the mechanism of gating in KtrB by nucleotide exchange in the KtrA ring remains elusive.

Here, we present a 2.5 Å cryo-EM structure of the inactive, ADP-bound KtrAB complex from *V. alginolyticus* that shows how the closed conformation is stabilized. Further, the N-termini of KtrB, which were unresolved in all previous structures, were identified at the interface of the two subunits and are shown to form a strong interaction network with KtrA and KtrB. By combining MD simulations, EPR data, and functional studies, we show that the intrinsically disordered N-termini in a lipid environment play a crucial role in the ATP-dependent activation of the system. Our data reveal that KtrAB from *V. alginolyticus* is regulated by an unforeseen complex allosteric mechanism, in which ATP triggers conformational changes within both subunits, including an interaction of the disordered N-termini with the membrane, thus enabling the opening of the intramembrane gate and $K^+$ flux.

## Results

### Structure of the natively assembled KtrAB complex

So far, all available structural information of the KtrAB system was based on a nonphysiological $KtrB_2A_8B_2$ complex assembly, where a second KtrB dimer is associated to the cytosolic side of the symmetric octameric KtrA ring[17,18,26]. In the ATP-bound state, this assembly could possibly hamper the complete activation of KtrAB. By saturating KtrB-containing membranes with a large excess of KtrA-containing cytosol prior to solubilisation, we optimized the purification protocol of full-length KtrAB from *V. alginolyticus*, resulting in the natively assembled $KtrB_2A_8$ complex (Supplementary Fig. 1). We obtained a cryo-EM map of the natively assembled complex at a final resolution of 2.8 Å (Fig. 1a and Supplementary Fig. 2a). Due to higher symmetry, an

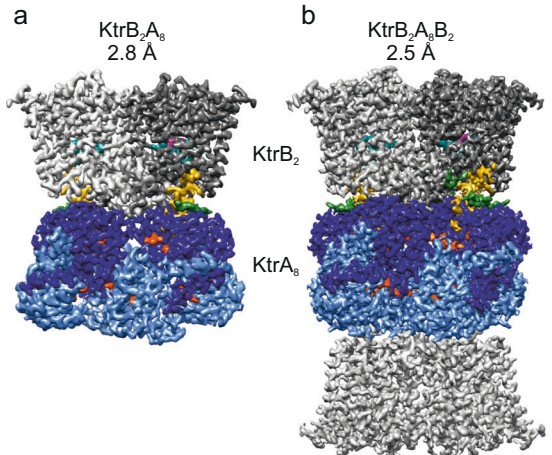

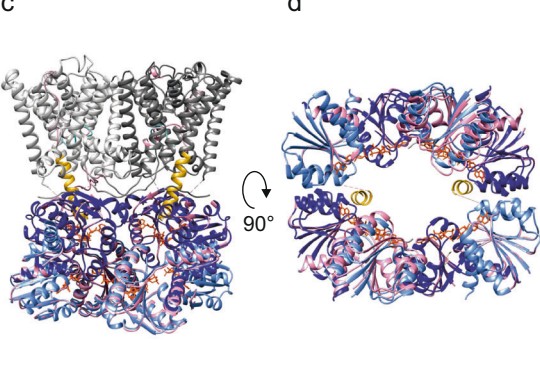

**Fig. 1 | High-resolution structure of KtrAB in its intact $KtrB_2A_8$ assembly compared to the nonphysiological $KtrB_2A_8B_2$ assembly.** Cryo-EM density maps of **a** $KtrB_2A_8$ with an overall resolution of 2.8 Å and **b** KtrAB in the nonphysiological $KtrB_2A_8B_2$ assembly with an overall resolution of 2.5 Å. **c** Side-view and **d** bottom-view of the overlay of both structures ($KtrB_2A_8B_2$ colored, $KtrB_2A_8$ pink). The assembly of an additional KtrB dimer to the free side of KtrA does not change the

overall structure of the KtrAB complex. Small rotations were identified in two KtrA subunits where the extra KtrB dimer binds, with no significant consequence on the remaining protein. KtrB: gray, D1M2 helix: goldenrod, intramembrane loop: dark cyan, R427: magenta, N-terminus of KtrB: green, KtrA: blue (navy blue, cornflower blue), ADP: orange. If not stated otherwise, coloring is maintained in all figures.

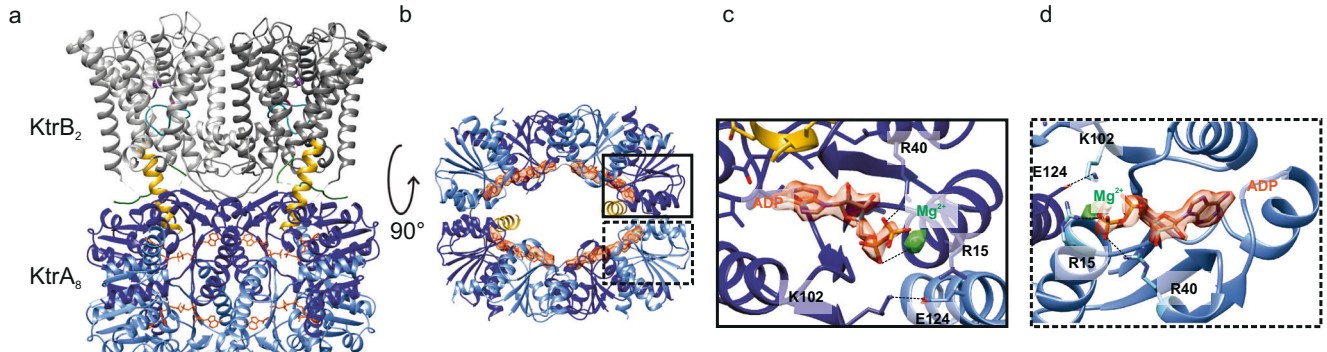

**Fig. 2 | Structure of the KtrAB complex in the ADP-bound conformation. a** Side-view of KtrAB with all features colored as described in Fig. 1, adopting the ADP-bound conformation. The D1M2 helices (goldenrod) extend into the KtrA ring. **b** View of KtrAB from the cytosol. The KtrA ring adopts the oval shape with a short axis of 6.3 nm and a long axis of 7.8 nm, as described previously for KtrAB in the ADP-bound conformation[18]. An ADP molecule is resolved in each KtrA subunit. **c** R40 in KtrA coordinates the β-phosphates of the ADP molecules. A non-protein density was assigned to a $Mg^{2+}$ (green) coordinated to the α-phosphate. **d** In alternating KtrA subunits, residue R15 coordinates the phosphate.

even higher resolution of 2.5 Å was observed for the complex in the $KtrB_2A_8B_2$ assembly, which was purified in a different attempt via ATP agarose (Fig. 1b and Supplementary Figs. 2b, 3). A comparison of the two structures showed that the assembly of the additional KtrB dimer to the cytosolic side of the KtrA ring has no significant effect on the structural arrangement of the protein (Fig. 1c, d). Therefore, the 2.5 Å cryo-EM map was used for further structural analysis.

## ADP-bound conformation of the KtrAB complex

Although all KtrAB preparations were done in the presence of ATP, the cryo-EM maps clearly display features that define the ADP-bound state, in particular the extended D1M2 helices and the oval-shaped KtrA ring (Fig. 2a)[18]. In agreement with this, the KtrA ring contains non-protein densities corresponding to ADP molecules bound to the N lobes of each of the eight KtrA subunits (Fig. 2b–d). The ADP coordination differs slightly from the X-ray structure of *Bs*KtrA (PDB 4J91) and the cryo-EM structure of *Bs*KtrAB (8K1S). While the positive charge of residue K102 (corresponding to *Va*KtrA, numbering +1 in *Bs*KtrA) is involved in ADP coordination in *Bs*KtrA (Supplementary Fig. 4), this is not observed for *Va*KtrA. Here, instead, residue R40 interacts with the negative charge of the β-phosphate. Furthermore, an additional small non-protein density, assigned as $Mg^{2+}$ due to the buffer composition and the coordination geometry, appears to be involved in the coordination of the α-phosphate (Fig. 2c). Interestingly, in every second KtrA subunit residue R15, which was described for the ADP coordination in *Bs*KtrA, is involved in the coordination of the ADP molecule, adding a positive charge to the binding site (Fig. 2d)[29]. E124, which has been shown to coordinate $Na^+$ in the recent ATP-bound structure[30], is far from the magnesium ion in the ADP-bound conformation. Instead, it interacts with K102 of the neighboring KtrA subunit. As expected, the ADP in the oval-shaped ring does not establish interactions with residues of the neighboring subunit, as observed in the $Na^+$ATP-bound, square-shaped ring.

Thus, despite preparing the sample in the presence of physiological ATP concentrations of up to 5 mM and 140 mM $Na^+$ or even purifying via ATP agarose, we obtained the structure of the ADP-bound KtrAB complex, which suggests that a decisive factor is missing to trigger the activated ATP-bound state.

## Exchange of ATP and ADP in the KtrA ring

For the isolated *Bs*KtrA ring a square-shaped conformation upon ATP binding was shown[28], and the ATP-bound *Bs*KtrAB complex was assembled by mixing ATP-bound KtrA with detergent-purified KtrB[17,30]. We assumed we could trigger a corresponding conformational change by adding ATP to the detergent-purified *Va*KtrAB complex. To

understand the lack of ATP binding, we first investigated the binding of ATP and ADP to isolated *Va*KtrA in the absence of KtrB. By using isothermal titration calorimetry (ITC), we confirmed the binding of both ADP and ATP to the isolated KtrA ring with low micromolar affinities ($K_D$s of 3.4 μM and 2.4 μM, respectively) (Fig. 3a)[18]. Furthermore, differential scanning fluorimetry (DSF) of the purified KtrA ring in the presence of both ATP and ADP showed an increase in the melting temperature (Fig. 3b–d), which indicates that both nucleotides bind to and stabilize the ring. However, the thermostability was significantly higher for ADP than for ATP (67 °C vs 55 °C, respectively), indicating a higher stability of KtrA in the ADP-bound state. Additionally, the presumably oval-shaped ADP-bound state appears to be stabilized at significantly lower concentrations (approx. factor 100) than the square-shaped ATP-bound state. This is also reflected in a DSF competition assay on KtrA alone, where, based on the shift in melting temperature, a shift from the ATP-bound state towards the ADP-bound state could be more efficiently reached than a shift from the ADP-bound state towards the ATP-bound state. While for ATP-bound KtrA the melting temperature for the ADP-bound state of 67 °C was recovered with 10 mM ADP, only a partial shift towards the ATP-bound state was reached when exchanging ADP with ATP (Fig. 3e). Last, we tested whether the binding of individual nucleotides actually stabilizes the predicted oval- and square-shaped conformations of the octameric ring, respectively. By stochastic analysis of the ring shape of 2D class averages from negatively stained KtrA particles, we confirmed that square-shaped and oval-shaped conformations are induced in the presence of ATP and ADP, respectively. Hence, we conclude that the ADP-bound, oval-shaped conformation is the preferred, more stable state of KtrA, but that in isolation, the ATP-bound conformation can be triggered by adding an excess of ATP. Consequently, the presence of the KtrB dimer must significantly alter the competition for nucleotide binding and the subsequent conformational change in the octameric ring (Supplementary Fig. 5).

## Stabilization of the closed ADP-bound state of the KtrAB complex

The high-resolution cryo-EM map of the ADP-bound complex makes it possible to determine the molecular basis of the stabilization of the inactive, closed conformation and possibly also to explain the changed kinetics of nucleotide binding. Decisive for ion translocation through the KtrB pore are the selectivity filter and the intramembrane loop, serving ion selectivity and gating. The cryo-EM map reveals a strong, well-defined density at a position corresponding to the S3 position of canonical TVGYG selectivity filters. The density is surrounded by the backbone carbonyls of residues V68 and T69 from D1, N183 and A184

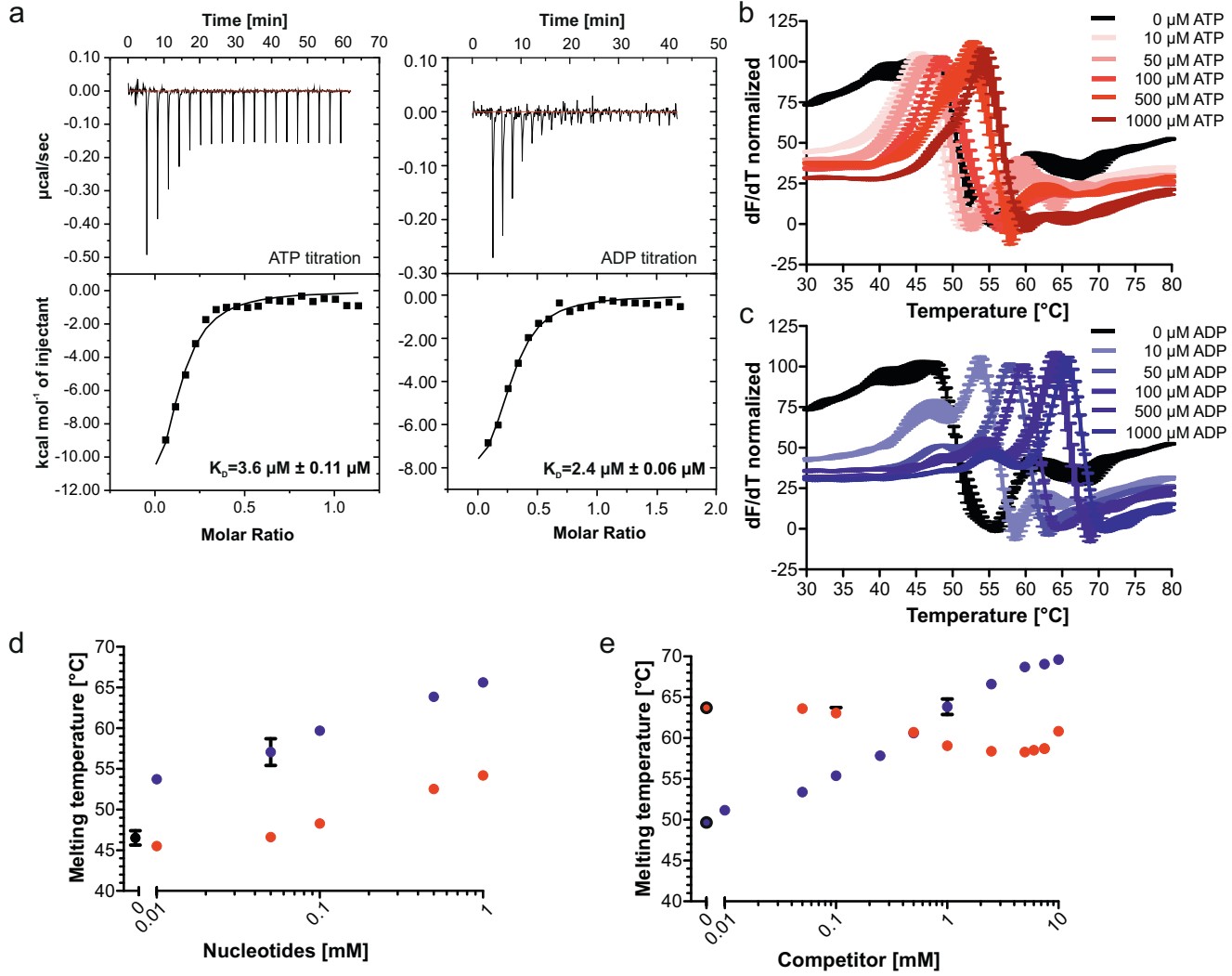

**Fig. 3 | Binding affinities of nucleotides to KtrA and nucleotide-dependent protein stabilization.** KtrA was purified via SEC and the affinities of nucleotides were determined using ITC. **a** The upper panel shows the raw heat exchange data of ATP and ADP binding to KtrA, respectively. The lower panels present the integrated injection heat pulses, normalized per mole of injection. The binding curves were fitted by a one-site binding model, resulting in the indicated dissociation constants ($K_D$). Respective $K_D$ values and SDs are noted for triplicates of each titration. **b, c** Thermostability of KtrA was investigated by differential scanning fluorimetry. Purified KtrA was mixed with different concentrations of ATP (red) or ADP (blue) and changes in SyproOrange fluorescence ($\lambda_{ex}$ 470 nm; $\lambda_{em}$ 555 nm) were detected over a temperature increase from 25 to 80 °C. The peak of the raw data's first derivative reflects the protein's melting point. **d** Melting temperatures of purified KtrA (black) and in the presence of different concentrations of ATP (red) or ADP (blue) (10 μM to 1000 μM). ADP binding induces higher thermal stability at significantly lower concentrations. **e** To test ATP/ADP displacement, competitive binding experiments were performed. 5 μM KtrA were pre-incubated with either 500 μM ATP (black-framed blue point) or 500 μM ADP (black-framed red point) and supplied with different concentrations of ADP (blue) or ATP (red), respectively. Displacement of both nucleotides was observed. Data points in d and e show the means and SDs of technical triplicates.

from D2, T288 and A289 from D3, and T400 and V401 from D4, resembling the coordination geometry of the first hydration shell of a potassium ion (Fig. 4a), similar to the coordination of dehydrated K[+] in other potassium channel[7]. This site was already described as an ion binding site in TrkAH and KtrAB[17,25], and consequently we assigned the density as a potassium ion. Two weaker densities are present above and below the S3 site, which would correspond to the S2 and S4 sites in a canonical K[+] selectivity filter. However, the surrounding residues do not appear to support the coordination of a dehydrated potassium ion and therefore the densities were assigned as water molecules (Fig. 4a). Nevertheless, unambiguous identification of ions in cryo-EM is problematic and it cannot be excluded that potassium ions at lower occupancy contribute to these densities[31]. The S1 site is clearly missing as the selectivity filter opens toward the outside; consequently, no extra density was found.

In agreement with the expected closed state, the intramembrane loop just below the selectivity filter blocks the ion permeation pathway together with the conserved residue R427 from D4M2 (Fig. 4b). Our structure shows that the intramembrane loop is indeed stabilized by interactions with the extended D1M2 helix, as previously speculated[18]. Calculating the radii along the pore of KtrB with HOLE[32] reveals a constriction at the intramembrane loop with a diameter as small as -0.2 Å, preventing the passage even of dehydrated K[+] (Fig. 4c). In particular, backbone interactions and hydrophobic interactions with residues S319 and T320 are formed. An additional steric barrier within the pore is established by the conserved residue K325 of the loop, which forms a salt bridge with the backbone carboxylate of the highly conserved C-terminal G455 of the neighboring KtrB that reaches into the pore (Fig. 4b). The substitution of K325 with an alanine led to a significant increase of the K[+] uptake velocities in whole-cell K[+] uptake

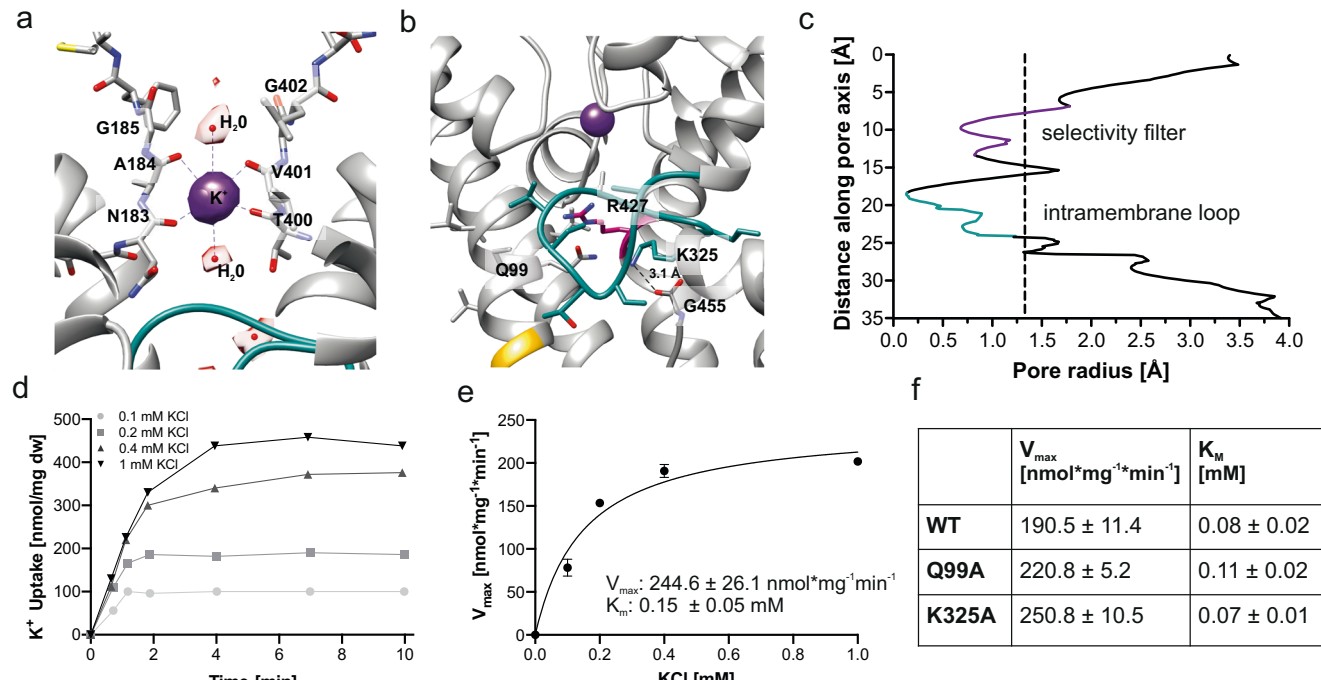

**Fig. 4 | Structural details of the selectivity filter and stabilization of the intramembrane loop of KtrB. a** Close-up view of the selectivity filter. Coordination of a potassium (purple density) in the selectivity filter (SF) at the S3 binding site by backbone carbonyl oxygens. Two additional non-protein densities in the SF were assigned as water molecules. D1 and D3 have been removed for better visualization. **b** Close-up view of the intramembrane loop (cyan). It is stabilized in the closed conformation mainly by interactions with the backbone of the extended D1M2 helix and the D4M2 helix, together with a salt bridge between K325 and the backbone carboxylate of the C-terminal G455 from the neighboring KtrB protomer. The conserved R427 (magenta) of helix D4M2, which is stabilized by Q99, forms an additional electrostatic barrier for potassium flux. **c** Calculated pore radii in a KtrB monomer along the central axis. For reference, the radius of a dehydrated K⁺ is shown as a dashed line. The pathway is clearly restricted by the intramembrane loop. Radii were calculated using HOLE[32]. **d** K⁺ uptake by *E. coli* LB2003 cells

producing KtrAB$_{K325A}$ was performed to determine uptake kinetics. After protein production, cells were depleted of K⁺ and, after 10 min incubation at room temperature, different K⁺ concentrations (0.1, 0.2, 0.4, and 1 mM, different shades of gray) were added. 1 ml samples were taken at different time points, and cells were separated from medium via centrifugation through silicone oil. Intracellular K⁺ concentration was determined by flame photometry. Uptake experiments were performed in biological triplicates with one representative graph shown. **e** Slopes from initial uptake velocities are plotted against the used KCl concentration with error bars resulting from the linear fitting. A Michaelis-Menten fit was performed to determine $V_{max}$ and $K_m$. The indicated error is of the individual fit. **f** Uptake experiments were also performed with the KtrAB$_{Q99A}$ variant and KtrAB WT for comparison (Supplementary Fig. 6). Average Michaelis-Menten kinetics for WT KtrAB, KtrAB$_{Q99A}$, and KtrAB$_{K325A}$ derived from biological triplicates. Both variants revealed increased $V_{max}$ compared to WT. The $K_m$ values remained similar.

experiments (Fig. 4f, Supplementary Fig. 6), indicating a destabilization of the closed pore. Further, R427 from D4M2 is stabilized inside the pore by residue Q99 from D1M2, providing another electrostatic barrier (Fig. 3b). Substitution of the glutamine with an alanine also destabilized the closed state, as this KtrAB variant similarly showed an increase in uptake velocity compared to the wild type (Fig. 4d–f). The apparent affinity ($K_m$) for both variants, however, did not significantly change (Fig. 4f). This is in good agreement with published uptake and transport experiments on variants of KtrB and KtrAB containing different mutations in the intramembrane loop, mainly showing an increase of K⁺ flux[23,30].

The extended helices D1M2 were previously suggested to be stabilized in the ADP-bound conformation by their interaction with the KtrA ring[18]. The present structure reveals the molecular details of these interactions. Although the helices are in close proximity to bound ADP molecules, there is no direct contact and KtrB does not contribute to ADP coordination, so nucleotide binding may not directly influence the conformation of the extended helices and vice versa. Instead, there are numerous interactions between the extended helices and the KtrA ring, most notably intersubunit salt bridges between the conserved R117 of the KtrB protomers and E38 of two KtrAs (Fig. 5a). Further, at the interface of the KtrA ring and the extended D1M2 helix of KtrB, an extended density lies flat on the KtrA ring, which we identified as the N-terminal residues R7 to D14 of KtrB (Fig. 5b, c). The N-terminus

includes a hydrophobic patch (residue V9-P13) that establishes several hydrophobic interactions with both the extended D1M2 helix and the KtrA ring. Furthermore, residue R7 forms a salt bridge with E64 of a KtrA protomer (Fig. 5c). The region between the N-terminus and the D1M1 helix, containing a motif of four positively charged residues (K16/R17/K19/K21), is not resolved in the map, indicating flexibility.

## The N-termini of KtrB are involved in the activation of KtrAB

The KtrB N-termini form a strong interaction network that appears to contribute substantially to the stabilization of the extended helices and consequently of the ADP-bound, closed KtrAB complex. To investigate this hypothesis structurally, a variant of KtrAB containing a deletion of KtrB's residues 2-19 was purified, supplemented with 1 mM ATP, and analysed by cryo-EM (Supplementary Fig. 7). The protein yield after SEC was comparable to the yields observed for KtrAB wild type, showing similar expression levels. A dataset of 58,000 particles contributed to a 3.6 Å-resolution cryo-EM map (Fig. 6a). While the density of the N-terminus was lacking as expected, the overall architecture of the complex exhibits all previously described features of the ADP-bound conformation (Fig. 6b, c), showing that, unexpectedly, the N-termini are not required for the stabilization of the complex in the closed state.

To investigate the behavior of the N-termini and get insights into possible functions, multiple all-atom MD simulations of KtrB and

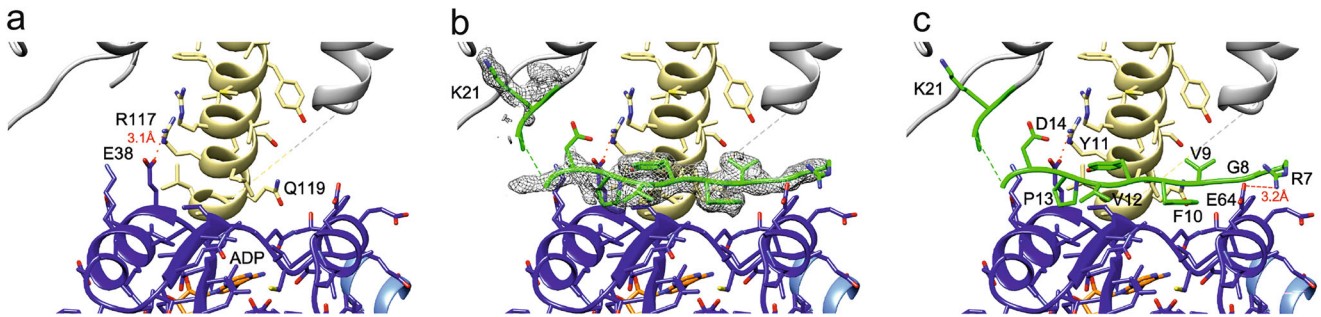

**Fig. 5 | Stabilization of extended D1M2 helices within KtrAs and identification of KtrB's unstructured N-termini at the interface of KtrA and KtrB. a** Extended helices of KtrB (yellow) are stabilized by several hydrophobic interactions and hydrogen bonds with KtrA (blue), including a salt bridge between R117 (KtrB) and E38 (KtrA) (distance 3.1 Å). **b** Cryo-EM density (gray mesh) and **c** model for the N-terminus of KtrB (green) at the interface of KtrB and KtrA, which is resolved from residue R7 to D14. R7 forms a stabilizing salt bridge with E64 from KtrA. A hydrophobic patch from V9 to P13 interacts with the extended helix and KtrA. A motif of three positively charged residues (K16, R17, K19) was not resolved, indicating high flexibility in this region.

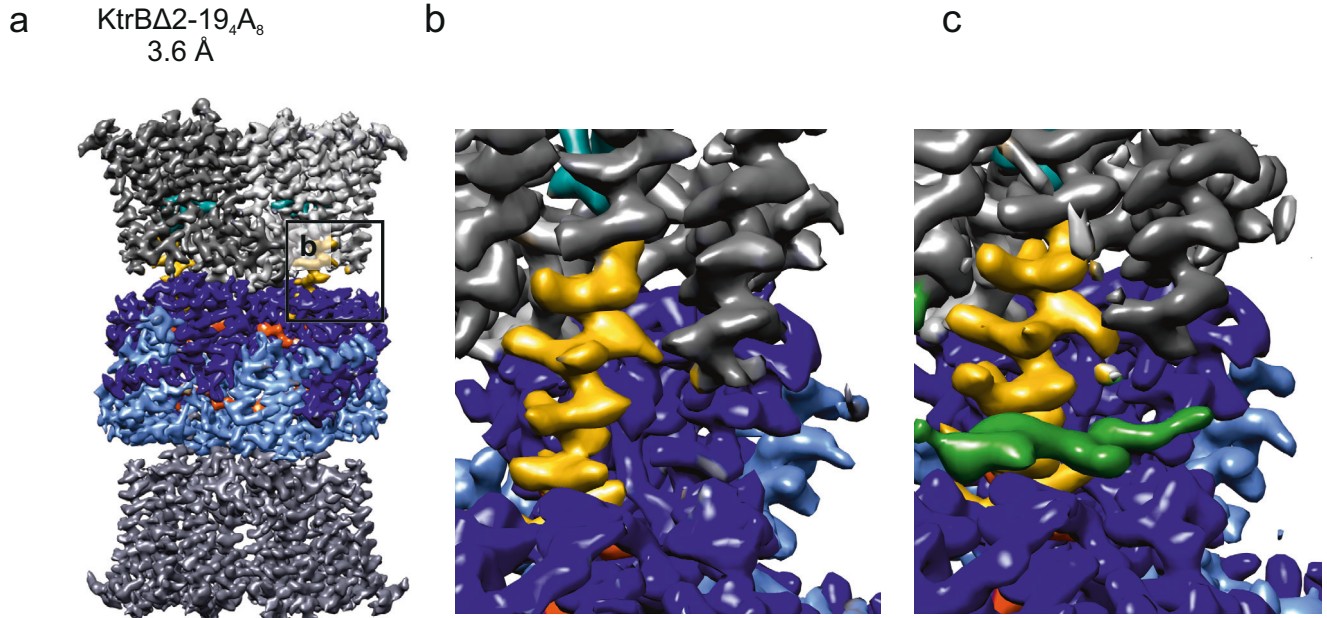

**Fig. 6 | Effect of N-terminus deletion on the KtrAB complex. a** Cryo-EM map at 3.6 Å resolution of the KtrBΔ2-19$_4$A$_8$ assembly. The absence of the N-terminus has no effect on the ADP-bound conformation of the system. **b** While the system adopts the ADP-bound conformation with extended D1M2 helices and an oval-shaped KtrA ring, density for the N-termini is lacking, as expected. **c** For comparison, a close-up view of full-length KtrAB low-pass filtered to the same resolution (3.6 Å) is shown, presenting the density of the N-terminus (green).

KtrAB, placed into a heterogeneous lipid bilayer comprising 45% POPE, 40% POPG and 15% cardiolipin were performed at three different temperatures (Fig. 7 and Suppl. Table 2). In the absence of KtrA, the N-termini of KtrB were found to be disordered (Fig. 7e) and to interact with the membrane (Fig. 7e–i), especially via their basic residues. At the higher temperatures, the apolar residues increasingly penetrate into the membrane, as seen visually and indicated by the distributions of the vertical distances $z$ of the N-terminal residues to the membrane surface (Fig. 7h, i). Distributions of the number of contacts between N-terminal residues and lipids follow the same trend (Fig. 7j). Supplementary Movie 1 details how, first, the basic residues interact with the negatively charged lipid head groups, followed by interaction of the hydrophobic patch with the acyl chains (Fig. 7d).

In contrast, in the presence of KtrA the N-termini remain in the position observed in the cryo-EM structure, i.e. mainly interacting with KtrA, indicating a stable interaction between the KtrB N-termini and KtrA (Fig. 7f). This is further supported by the analysis of the fraction of native contacts between the KtrB N-termini and KtrA, which remain high even at high temperatures (Fig. 7g). Furthermore, analysis of the N-terminus length as a measure of conformational change of this region shows that the N-termini keep the same conformation in the KtrAB simulations, while with only KtrB the N-termini explore a broader distribution during the simulations (Fig. 7h). Similar conclusions can be drawn by the distance of the N-termini from the membrane, which remains the same at all temperatures while the number of interactions stays low compared to the KtrB dimer (Fig. 7i, j). The set of simulations shows that both interacting with the lipid bilayer and with KtrA is favorable for the intrinsically disordered N-termini.

During the simulations, we did not see any transition from N-termini interacting with KtrA to a full interaction with the membrane. However, in the higher-temperature simulations of KtrAB, we clearly observe a less stable interaction between the N-termini and KtrA and at the same time transient interaction of the N-termini with the membrane (Fig. 7g–j). Therefore, it seems that there is a significant

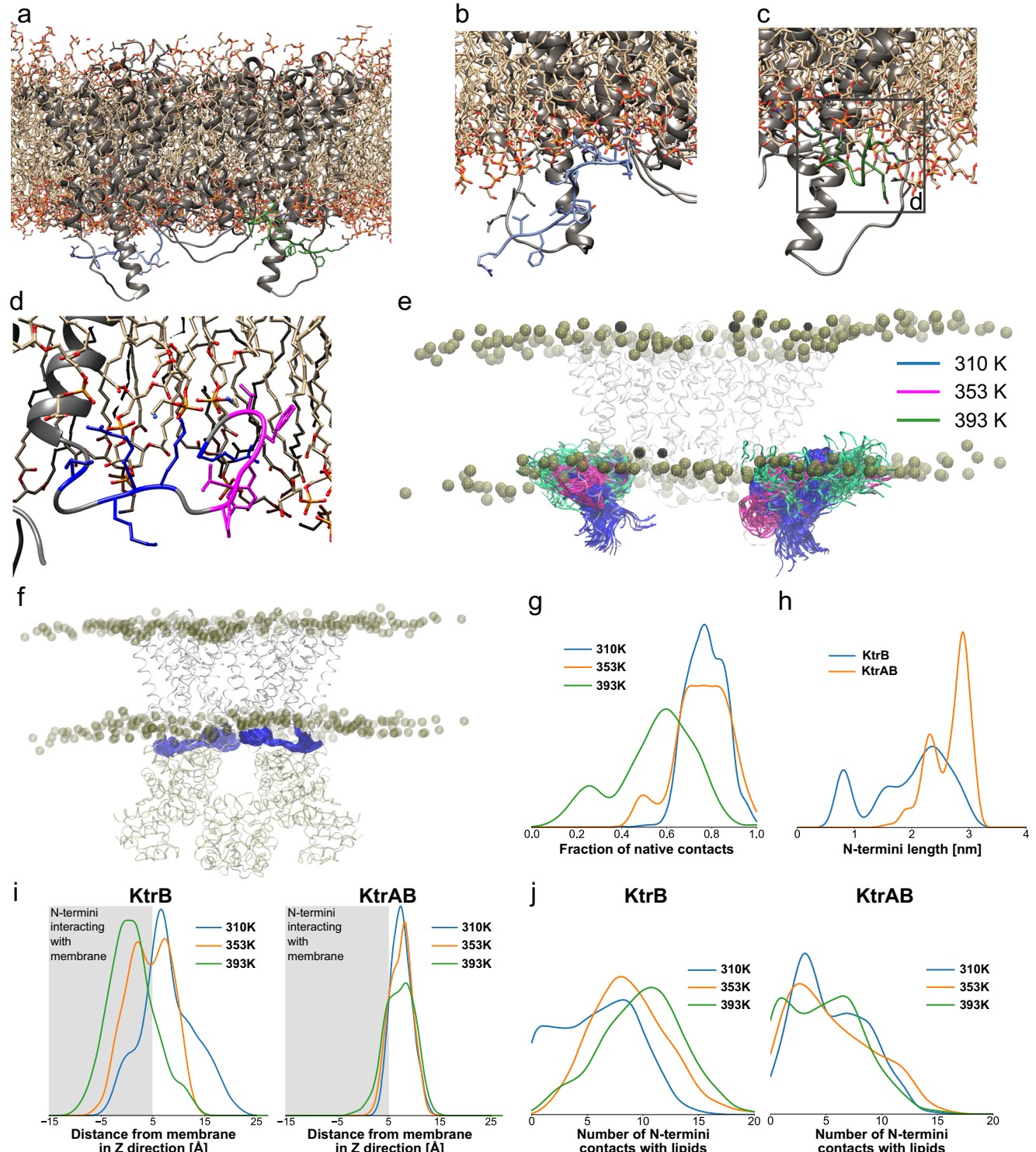

**Fig. 7 | All-atom MD simulations of KtrB dimer and the KtrAB complex in a lipid environment. a** Snapshot of the membrane and the KtrB dimer after an initial equilibration of the system. The disordered N-termini (blue and green) hover below the membrane. **b** After 1 μs of MD simulation at 310 K, the N-terminus of one protomer (blue) interacts with the membrane surface, primarily via its positively charged residues. **c, d** The N-terminus of the other protomer (green) shows a strong interaction with the lipid membrane after 1000 ns of simulation time at 310 K, with the hydrophobic patch (pink in d) interacting with the acyl chains while the basic residues (blue in d) interact with negatively charged head groups. **e** Dynamics of the N-termini during the MD simulation of KtrB at different temperatures. Snapshots of the N-termini (blue) taken every 1 ns from the 1 μs simulation after superimposition of the protein core. **f** Dynamics of KtrB's N-termini (blue) in the KtrAB complex during MD simulations at 310 K. Snapshots of the

N-termini taken every 1 ns from the 1 μs simulation after superimposition of the protein. **g** The fraction of native contacts between KtrA and the N-termini of KtrB remains high in the KtrAB complex even at elevated temperatures. **h** The length of KtrB's N-termini varies in the KtrB dimer, showing their unstructured nature. In the KtrAB complex, the length is stable as the termini remain in the pocket between the subunits. **i** Distributions of the distance of the KtrB N-termini from the membrane surface in the z direction (in Å) in the MD simulations of the KtrB dimer and the full KtrAB complex. The membrane is indicated by the gray shading, z = 0 corresponding to the average phosphate positions. **j** Distribution of the number of contacts formed between lipids and N-terminal residues of KtrB in the MD simulations of the KtrB dimer and the full KtrAB complex. The time series of the results of panels **g**–**j** are shown in Supplementary Fig. 8.

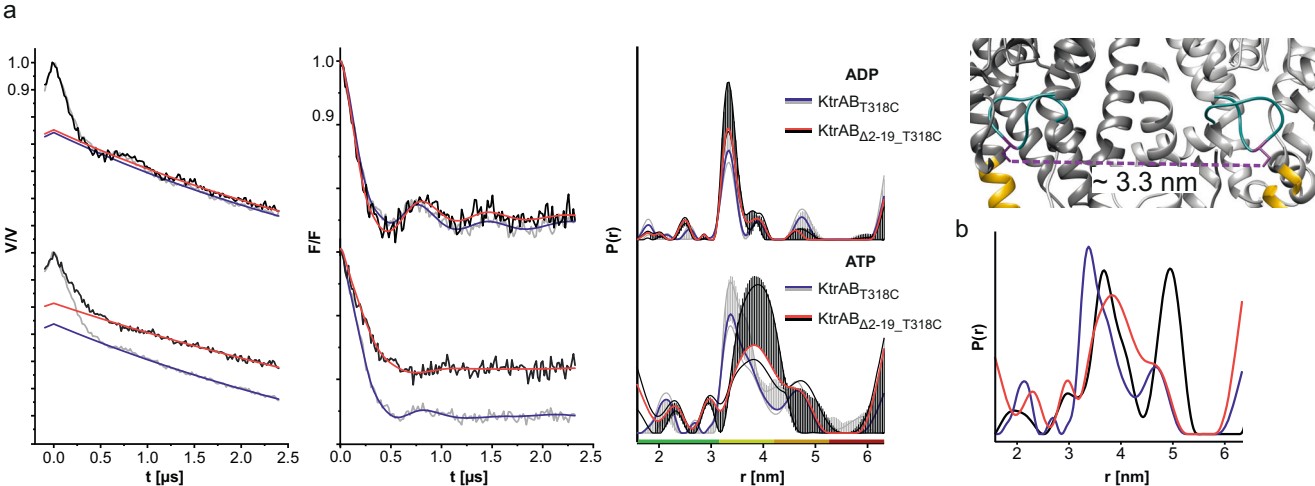

**Fig. 8 | Dependency of the intramembrane loop flexibility of KtrAB on nucleotides and KtrB N-termini. a** DEER measurements of spin-labeled natively assembled KtrAB$_{T318C}$ and KtrAB$_{\Delta 2\text{-}19\_T318C}$ in the presence of 10 mM ADP (blue) or 10 mM ATP (red) reconstituted in *E. coli* polar lipid liposomes with a lipid:protein ratio of 1:10. Left panels: Experimental raw data V(t) with fitted background function of individual measurements; middle panels: Background-corrected dipolar evolution functions F(t); right panels: Interspin distance distributions P(r) obtained by Tikhonov regularization. Light gray and black striped areas, respectively, represent the full variation of possible distance distributions. The lower and upper error estimates (light gray and black lines, respectively) represent the respective mean values plus and minus twice the standard deviation. On the x-axis, the green range corresponds to a reliable shape of the distribution, the yellow range to

reliable mean distance and width, the orange range to reliable mean distance (but not width), and the red range to recognition of a long-distance contribution that cannot be quantified. DeerAnalysis2022 was used for the analysis. Distances are measured between the labeled residues in both protomers, the indicated distance in the model is the mean distance resulting from a rotamer library analysis using MMM2020.1. While in the presence of ADP the interspin distances resemble each other, the gating loops must be significantly delocalized in KtrAB$_{\Delta 2\text{-}19\_T318C}$ in the presence of ATP when compared to full-length KtrAB$_{T318C}$. **b** Comparison of distance distributions from KtrB$_{T318C}$ (black), KtrAB$_{T318C}$ + ATP (blue), and KtrAB$_{\Delta 2\text{-}19\_T318C}$ + ATP (red). The corresponding DEER measurement of KtrB$_{T318C}$ is shown in Supplementary Fig. 10.

but not insurmountable energetic barrier between these two states. We speculated that this transition might be facilitated by partial binding of ATP to KtrA and that consequently an allosteric mechanism of nucleotide-triggered conformational changes and relocation of the N-termini towards the membrane is required for achieving the active, open state.

To test this hypothesis, we first obtained cryo-EM data of KtrAB reconstituted into nanodiscs in the presence of ATP. However, KtrAB still clearly resembled the ADP-bound state (Supplementary Fig. 9). A possible explanation is that the provided lipid surface in the nanodisc was too narrow for the N-termini to bind. Therefore, we instead concentrated on the flexibility of the gating intramembrane loop as another key element of the hypothesis. We probed the loop motions by pulsed EPR measurements under the assumption that the flexibility changes in the active versus the inactive state. Variant KtrB$_{T318C}$ has been successfully used to analyse the flexibility of the intramembrane loop by EPR spectroscopy in the absence of KtrA[24]. Therefore, a spin-labeled KtrAB$_{T318C}$ variant was reconstituted in liposomes consisting of *E. coli* polar lipids in the presence of 10 mM ATP or ADP and a K$^+$ gradient over the membrane (Fig. 8a), and the distance distribution between the two spin labels (one per protomer) was determined. In the presence of ADP, a clear oscillation in the experimental raw data and the background-corrected dipolar evolution function (F(t)) was visible, resulting in a narrow distance distribution centered at 3.3 nm. We assume that this distance distribution represents the immobilized intramembrane loop in the closed state as it is in good agreement with the distance distribution predicted for the ADP-bound cryo-EM model using MMM2020.1 (Fig. 8a and Supplementary Fig. 10a)[33,34]. In the presence of ATP, the mean distance distribution was significantly broadened towards longer distances. We conclude that the broadening reflects a more flexible gate that adopts different states to facilitate potassium ion flux. Consequently, in the liposome-based system, the addition of ATP is sufficient to trigger conformational changes.

To further examine the role of the N-termini, the same measurements were performed without the N-terminal residues 2-19 of KtrB. Interestingly, while for ADP a nearly identical distance distribution was observed compared to full-length KtrAB, which is in line with the cryo-EM model of KtrAB$_{\Delta 2\text{-}19}$, in the presence of ATP the distance distribution reporting on the intramembrane loop motions was further broadened and shifted towards even longer distances than the full-length complex (Fig. 8a). In fact, the distance distribution reflects the distance distribution of spin-labeled KtrB$_{T318C}$, hence in the absence of regulatory KtrA, centered at 3.7 nm (Fig. 8b and Supplementary Fig. 10b), and suggests a significant delocalization of the gating loop when compared to full-length KtrAB. Overall, the EPR measurements give another indication that the interaction between the N-termini and the membrane in combination with nucleotide-triggered conformational changes in KtrA is crucial for the transition between the non-conducting and conducting states of the KtrAB channel.

If the latter hypothesis is correct, a deletion of KtrBs' N-termini should significantly impact KtrAB's activity. To test this, we determined the whole-cell uptake activity of the KtrAB complex with different mutations at the KtrB N-terminus (Fig. 9 and Supplementary Fig. 11). Deletion of the N-terminal peptide drastically decreased K$^+$ uptake, with V$_{max}$ dropping from 197 nmol*mg$^{-1}$*min$^{-1}$ for wildtype KtrAB to 53 nmol*mg$^{-1}$*min$^{-1}$, confirming the crucial role of the N-termini in the activation of the channel (Fig. 9a). Of note, the V$_{max}$ of the KtrAB complex without the KtrB N-terminal IDRs resembles the one of the KtrB dimer in the absence of KtrA, which supports the hypothesis that the gating region is indeed decoupled from the regulatory KtrA subunit, while the attached KtrA still significantly lowers the K$_M$ as described by Mikuševič et al. (2019) (Fig. 9b). To identify individual residues responsible for the impaired K$^+$ uptake, we mutated residues with different chemical properties and determined kinetic parameters (V$_{max}$ and K$_m$) for all variants (Fig. 9 and Supplementary Fig. 11). Partial mutation of the hydrophobic patch to serines (F10S/Y11S/V12S) did not significantly affect K$^+$ uptake, while the substitution

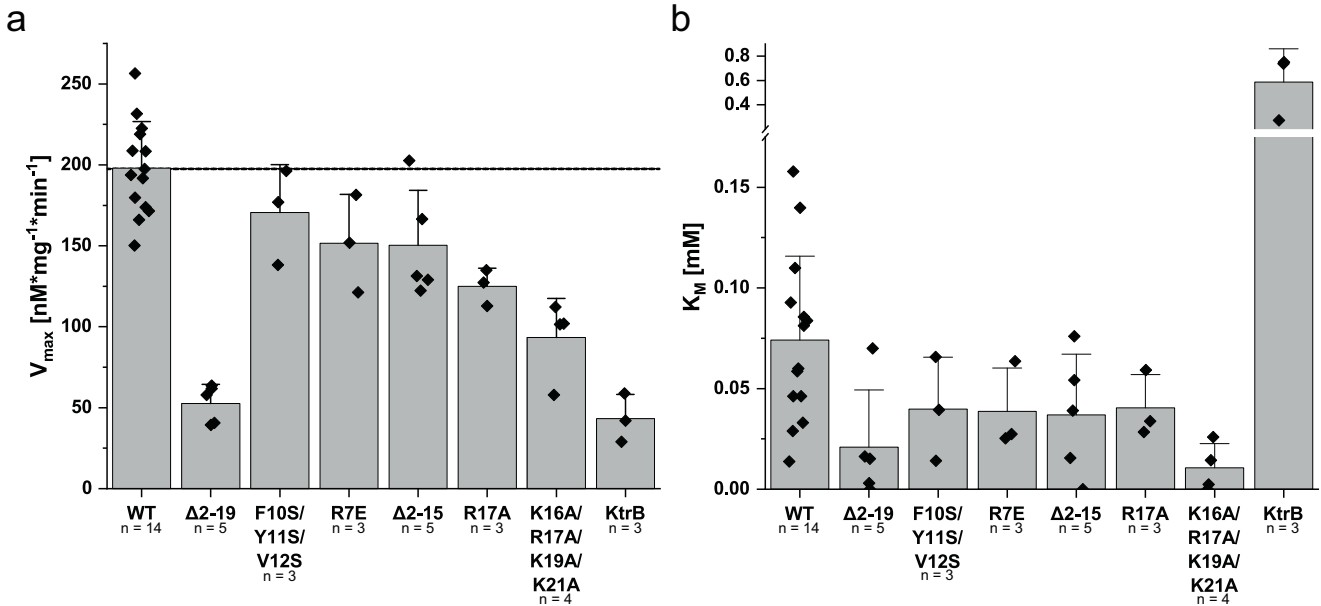

**Fig. 9 | Deletion and mutations of the KtrB N-terminus affect K⁺ uptake by the KtrAB complex. a, b** $V_{max}$ (**a**) and $K_M$ (**b**) for KtrB IDR variants. Uptake experiments were performed as described above. *E. coli* LB2003 cells produced KtrAB WT and variants thereof, as well as KtrB in the absence of KtrA. K⁺-depleted cells were prepared, and K⁺ uptake was measured in the presence of 0.1, 0.2, 0.4, and 1 mM KCl. All measurements were performed several times as indicated in the figure ($n = x$) and the Michaelis-Menten kinetics were determined using initial uptake velocities. Data are presented as a scatter plot graph with gray bars representing mean values +/− SDs. Exemplary uptake data are shown in Supplementary Fig. 11.

of R7 with a glutamate slightly reduced $V_{max}$ compared to wildtype. However, neutralization of all residues in the positively charged cluster of the IDR (K16A/R17A/K19A/K21A) reduced the $V_{max}$ by half. This critical role of the positive charge concentrated on the IDR is in line with the MD simulations, where the positively charged cluster appeared to pull the IDR to the membrane, forming strong interactions with the lipids. The MD simulations, EPR data, and K⁺ uptake assays together show that the KtrB N-terminal IDR acts as a signal transducer between the KtrA and KtrB subunits that is essential for ATP-dependent channel opening.

## Discussion

By combining structural, functional, and EPR data, and MD simulations, we revealed how the ADP-bound conformation of KtrAB from *V. alginolyticus* is stabilized. We found that the gating for K⁺ translocation does not only depend on ligand binding to the RCK domain. In addition, an allosteric mechanism is required that includes the interaction of KtrB's disordered N-termini with the membrane and conformational rearrangements within KtrB as a result of nucleotide-triggered conformational changes in KtrA. Based on these data we propose a more detailed gating mechanism (Fig. 10): In the preferred ADP-bound conformation, the RCK ring adopts an oval-shaped conformation and the D1M2 helices are extended, reaching into the KtrA ring. The N-termini of KtrB lie flat on the KtrA ring and the intramembrane loop blocks the pore for potassium flux.

Activation of the KtrAB system occurs as a consequence of osmotic stress, which is characterized by changes in ionic strength, ΔpH, lipid composition, membrane potential, turgor pressure[35], and an increase in the internal ATP concentration[36]. We hypothesize that an increased binding of ATP to KtrAB within the cell destabilizes the inactive conformation of KtrAB, allowing a displacement of the KtrB N-termini towards the membrane where they establish electrostatic contacts with the negatively charged phospholipid headgroups. Subsequently, a hydrophobic interaction of the hydrophobic patch in the N-termini with the acyl chains locks this interaction (Fig. 7d). In

consequence, this triggers conformational changes in the adjacent transmembrane helices of KtrB, especially of the extended D1M2 helices. Only after a relocation or breaking of the D1M2 helices[18], the KtrA ring can adopt the square-shaped conformation upon binding of ATP and Na⁺ to all eight binding sites. While in particular the movements of D1M2 helices break the interactions with the intramembrane loops, conformational changes in the KtrA ring affect the repositioning of KtrB's C-termini, as they form a lateral contact with KtrA. The C-termini in turn pull on the loops, which leads to a more flexible intramembrane loop of the other protomer. Consequently, potassium ions are taken up. The influx of K⁺ results in an increase of intracellular positive charges, shielding the negatively charged membrane surface. This leads to the repulsion of the N-termini from the membrane, which initiates the inactivation of the system to avoid uncontrolled K⁺ influx. The ATP concentration drops, and ATP gets exchanged by ADP in the KtrA ring, which consequently returns to the oval-shaped conformation, allowing the D1M2 helices to extend and re-establish interactions with KtrA. Meanwhile, the KtrB N-termini reorganize at the interface of the two subunits.

Interestingly, KtrAB from *B. subtilis* appears to behave differently. Here, the ATP-bound state could be obtained by only adding ATP and Na⁺ to the assembled KtrAB complex, with Na⁺ being coordinated between the γ-phosphates of two bound ATP molecules which stabilizes the square-shaped KtrA ring[30]. However, the intramembrane loop is still blocking the pore and it remains unclear whether a stabilized open conformation actually exists or whether unlocking of the loop is sufficient for activation. In contrast to our structural data, the N-terminus of *Bs*KtrB was not resolved in either the ATP- or ADP-bound conformation, and a sequence alignment shows that *Bs*KtrB indeed lacks the long N-terminus with a hydrophobic patch as in *Va*KtrB, raising the question of how conserved the regulatory mechanism proposed here, including N-termini, is.

An alignment of KtrB's N-termini from different species, from the starting methionine to the first transmembrane helix, reveals that the N-termini of KtrB from Gram-positive species are generally shorter

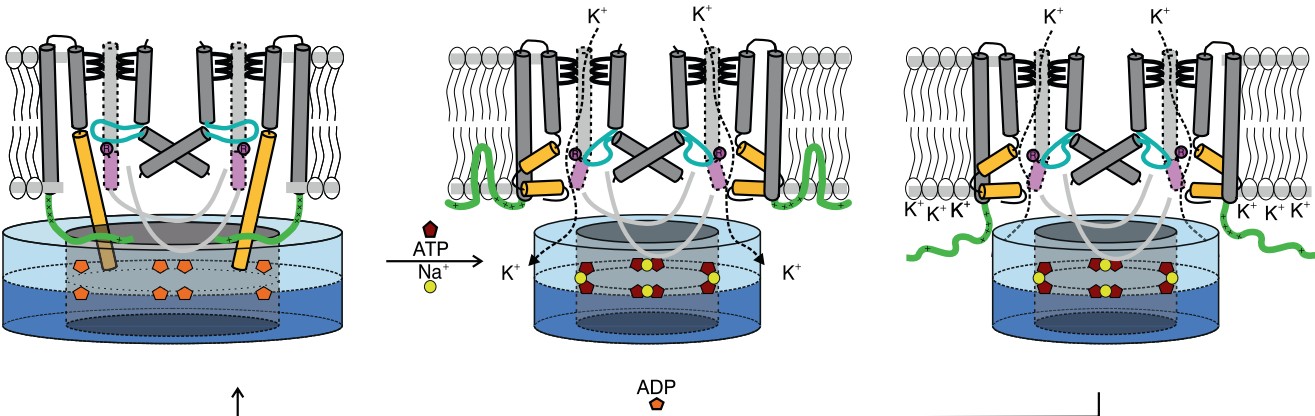

**Fig. 10 | Suggested gating mechanism of the KtrAB complex.** Cartoon representing an allosteric mechanism in which the interaction of KtrB's N-termini with the membrane, together with ATP- and Na$^+$-triggered conformational changes within KtrA, induce the activation of the KtrAB system. In the ADP-bound conformation (left, ADP in orange), the RCK ring (blue) has an oval shape, and the D1M2 helices are extended (gold), reaching into the KtrA ring. The N-termini of KtrB (green) lie flat on the KtrA ring, and the intramembrane loop (cyan) blocks potassium flux. Middle: Upon hyperosmotic stress, which goes along with an increased ATP concentration, the ADP-bound conformation gets destabilized by nucleotide exchange, and the N-termini relocate towards the membrane, establishing electrostatic contacts with the negatively charged phospholipid headgroups via several positive charges. Subsequently, a hydrophobic interaction of the hydrophobic patch with the acyl chains locks the interaction. This induces conformational rearrangements in KtrB: the D1M2 helices break, and helices D4M2 (pink) containing R427 relocate away from the pore. In parallel, in KtrA, further replacement of ADP by ATP (dark red) and Na$^+$ results in a square-shaped KtrA ring, pulling open the unlocked gate. Subsequent potassium uptake increases the intracellular positive charge (right), which locally shields the negatively charged membrane surface, leading to a repulsion of the N-termini. This is the first step of restoring the inactive, ADP-bound conformation to avoid K$^+$-mediated cytotoxic processes by unhindered uptake. Adapted from ref. 18.

than those from Gram-negative species (Supplementary Fig. 12). Despite a generally low conservation of KtrB's N-termini, the hydrophobic patch and the cluster of basic residues behind that patch are conserved in all of the compared sequences from Gram-negative bacteria. The shorter N-termini of KtrB from Gram-positive bacteria retain clusters of basic residues but lack the hydrophobic patch. We hypothesize that the attachment of the hydrophobic part of the N-terminus to KtrA and thus the interaction with the plasma membrane serves as an alternative input for modulating KtrAB's activity, particularly in Gram-negative bacteria. In most Gram-positive and some Gram-negative bacteria, cyclic di-AMP binding to KtrA grants the efficient inactivation of KtrAB at increased K$^+$ levels[37,38]. In organisms without this second messenger regulating potassium homeostasis at various levels[39], another regulatory input might be required. We showed here that, while the binding affinities for individual nucleotides to KtrA appear to be similar, the ADP-bound state is preferred over the ATP-bound state being stably formed even at lower concentrations. An explanation could be that the square-shaped, ATP-bound ring can only be adopted when all eight binding sites have bound ATP to coordinate a Na$^+$ between each two ATP molecules. The interaction of the N-termini with negatively charged phospholipid headgroups in the plasma membrane as suggested here is an attractive additive regulatory mechanism because at activating conditions, i.e., hyperosmotic stress, the concentration of negatively charged cardiolipin increases. At the same time, the ATP concentration transiently increases at the expense of ADP[35]. Hence, two physiological adaptations add up as direct stimuli for the activation of KtrAB when it is needed the most. In addition, detachment of the N-termini from the membrane could be an efficient and fast mechanism to inactivate KtrAB when sufficient K$^+$ has been taken up, while the ATP concentration might still be increased.

Of note, the importance of protein-lipid interactions for the regulation of osmo-protective transporters like OpuA, ProP and BetP, which are activated in the second phase of osmoadaptation, has been known for several years[40–42]. These proteins contain a domain that senses the ionic strength. In its inactive state it is bound to the negatively charged phospholipids. A detachment from the membrane through higher ionic strength leads to the activation of the transporters. Similar to the N-termini of KtrB, positively charged residues within the sensor domains of OpuA were shown to play a crucial role in the interaction with the membrane[43]. As these transport systems are activated when KtrAB is inactivated, an inverse mechanism is suggested. While the transporters are active when the interaction with the membrane is weakened, we hypothesize that KtrAB is activated upon strong interaction of the N-termini with the membrane.

Generally, modulation of ion channels by lipid interactions comes more and more into focus: The N-termini of the ancestral KcsA channel from *S. lividans* were shown to interact with the membrane and to play an important role in channel gating[44]. SthK, a bacterial pacemaker channel homolog, binds anionic lipids, which unlock the channel[45]. Lastly, human cation channel TRPV4, which, among others also serves osmoregulation, modulates channel activity in a lipid-dependent manner via its very long N-terminal IDR[46].

KtrAB and its homologs have been named pathogenicity factors in many bacterial pathogens[47–49]. Hence, it remains important to identify the active, open state of KtrAB to obtain a detailed understanding of the full gating mechanism, which could help to identify new effective inhibitory drugs[50]. For KtrAB homologs with the intrinsically disordered N-terminus providing a larger lipid surface might be key.

## Methods

### Expression and purification of native assembled KtrAB

For the purification of natively assembled KtrAB variants, the two subunits KtrA and KtrB were individually produced. Expression of tagless *ktrB* was induced with 0.02% L-arabinose in *E. coli* LB2003 cells[51] cultured in 12 l K3 minimal medium[51,52] supplemented with 100 µg/ml ampicillin. Cells were incubated aerobically at 37 °C for 5 h until $OD_{600}$ of approx. 1.2 was observed. Cells were harvested at 5500 *g* for 10 min and resuspended in buffer S (50 mM Tris-HCl pH 7.5, 420 mM NaCl, 180 mM KCl) to a final $OD_{600}$ of 100 and prior to cell disruption (Stansted, pressure cell homogenizer) the suspension was supplemented with DNaseI, 1 mM EDTA, 100 µg/ml phenylmethylsulfonyl fluoride (PMSF) and 300 µg/ml benzamidine. Cell

disruption was performed at 1 bar, and after low-speed centrifugation at 15,000 g for 15 min to remove unbroken cells, KtrB-containing membranes were isolated by ultracentrifugation at 130,000 g overnight at 4 °C.

KtrA containing a C-terminal His tag (KtrAC3H) was cloned using FX-cloning[53] and the gene was expressed in *E. coli* C43(DE3)Δ*acrAB* cells aerobically in 1 l LB media with 100 µg/ml at 37 °C. Protein expression was induced with 0.02% arabinose at an $OD_{600}$ of 0.8. After 1.5 h of aerobic growth at 37 °C, the cells were harvested at an $OD_{600}$ of at least 3 and the cells were disrupted as described above for KtrB. After ultracentrifugation at 130,000 × g for 1 h the KtrA-containing supernatant was mixed with the homogenized KtrB-containing membrane pellet to a final membrane concentration of 100 mg/ml. After mixing, the sample was solubilised with 1% DDM for 1 h at 4 °C under gentle agitation. Unsolubilized particles were removed by ultracentrifugation at 130,000 × g for 30 min and subsequently the supernatant was incubated with pre-equilibrated $Ni^{2+}$-NTA agarose for 2 h. Then, the $Ni^{2+}$-NTA resin was washed with 50 column volumes of buffer W (20 mM Tris-HCl pH 7.5, 140 mM NaCl, 60 mM KCl, 0.04% DDM) supplemented with 50 mM imidazole to remove any unspecific protein. For elution, the protein was cleaved off the $Ni^{2+}$-NTA beads with 3 C protease. For this, 3 C protease was used at a ratio of 1 mg of protease per litre *E. coli* C43(DE3)Δ*acrAB* pBKtrAC3H cell culture with an $OD_{600}$ of 3. The protease was resuspended in 2x CV buffer W containing 0.04% DDM and mixed with $Ni^{2+}$-NTA overnight at 4 °C under constant rotation. Elution fractions were collected, and the agarose beads were washed twice with 1x CV buffer W containing 0.04% DDM. The protein-containing fractions were combined and further purified by size exclusion chromatography using a Superdex200Increase 10/300 GL column (GE Healthcare) previously equilibrated with either cryo-EM buffer (20 mM Tris-HCl pH 7.5, 70 mM NaCl, 30 mM KCl, 0.025% DDM) or buffer W, depending on the purpose.

### Purification of KtrAB via ATP agarose

To purify KtrAB in the ATP-bound state, the complex was further purified via agarose immobilized γ-Aminophenyl-ATP (C10-spacer) (Jena Bioscience). For this purpose, the buffer was exchanged to buffer W with 0.04% DDM after elution of $Ni^{2+}$-NTA agarose, using Zeba™ Spin Desalting Columns (Thermo Fisher) to remove the imidazole. Afterwards, the protein was mixed with pre-equilibrated ATP agarose. Here, 4 mg $KtrA_8B_2$ were mixed with 200 µl of pre-equilibrated ATP agarose. The agarose has an ATP substitution of 20 µmol to 22 µmol ATP/ml slurry. The mixture was incubated overnight at 4 °C under gentle agitation. Subsequently, it was transferred to a chromatography column, and the flowthrough was collected. The agarose beads were washed with 50x CV buffer W + 0.04% DDM to remove unbound protein. Finally, the protein was eluted with buffer W + 0.04% DDM supplemented with 30 mM ATP and 2 mM $MgCl_2$ and further purified by size exclusion chromatography using Superdex200Increase 10/300 GL column (GE Healthcare). During SEC, the ATP concentration in the cryo-EM buffer was reduced to 1 mM ATP.

### Reconstitution into MSP2N2 nanodiscs

For the reconstitution of $KtrA_8B_2$ into nanodiscs, *E. coli* polar lipids in chloroform were dried and afterwards hydrated in buffer W (20 mM Tris-HCl pH 7.5, 140 mM NaCl, 60 mM KCl) with a final concentration of 5 mg/ml and solubilised in 1% DDM for 3-4 h. Afterwards, detergent-solubilised native KtrAB was incubated with lipids in a molar ratio of 1:100 or 1:300 (KtrAB complex: lipids). After 30 min incubation at room temperature and constantly mixing, MSP2N2 (1 mg/ml) was added in a molar ratio of 1:2.5:100 (KtrAB:MSP2N2:lipids). The mixture was further incubated for 30 min at 4 °C. Subsequently, BioBeads were added in three steps, each 40 mg/ml wet weight and incubated for

30 min, 45 min and overnight at 4 °C, respectively. BioBeads were removed, and samples were concentrated and filtered for SEC as described before.

### Spin-labeling of KtrAB variants

For spin-labeling, 5 mM β-mercaptoethanol were added to the solubilisation mixture. After binding to $Ni^{2+}$-NTA agarose, the beads were washed with 50x CV of buffer W containing 0.04% DDM, 50 mM imidazole and 5 mM β-mercaptoethanol, followed by an additional wash step with 15x CV ice-cold and degassed buffer W with 0.04% DDM to remove β-mercaptoethanol. For spin-labeling, 2x CV buffer W supplemented with 0.04% DDM and 1 mM MTSSL were added to the beads and after 1 CV of buffer passed through the beads, the column was closed and incubated with the remaining buffer overnight at 4 °C. Subsequently, the $Ni^{2+}$-NTA agarose beads were extensively washed with 30 CV buffer W with 0.04% DDM to remove excessive MTSSL. The protein was finally eluted with buffer W containing 0.04% DDM and 500 mM imidazole, followed by SEC. The samples were measured either detergent-solubilised after SEC concentrated to -10 mg/ml or reconstituted into liposomes with a final lipid concentration of -100 mg/ml.

### Reconstitution into liposomes

Spin-labeled proteins were reconstituted into liposomes containing *E. coli* polar lipids (prepared from Avanti total lipid extract) 1:10 protein to lipid ratio (w/w). Liposomes were prepared as described before[24]. The liposomes were diluted with buffer I (20 mM Tris-HCl pH 7.5, 195 mM KCl, 5 mM NaCl, 1 mM $MgCl_2$) to 4 mg/ml and titrated with Triton X-100 (Sigma). The detergent-destabilized liposomes were mixed with spin-labeled protein and incubated for 30 min at room temperature under gentle agitation. Polystyrene beads (Bio-Beads SM2) were added at a wet weight of 40 mg/ml, and the sample was incubated for further 15 min at room temperature. Subsequently, fresh BioBeads SM2 (40 mg/ml) were added four times with incubations at 4 °C of 15 min, 30 min and overnight. After o/n incubation 80 mg/ml of fresh BioBeads SM2 were added and incubated for 1 h. The beads were removed and the proteoliposomes were collected by ultracentrifugation at 450,000 × g for 30 min. Afterwards they were resuspended in the same buffer to have a lipid concentration of 10 mg/ml. To incorporate ADP or ATP into the liposomes, 10 mM ADP and 10 mM ATP, respectively, were added to the samples and three cycles of freeze and thaw were performed. Subsequently, the samples were extruded 11 times through a 400 nm filter and centrifuged for 30 min at 450,000 × g. The pellet was washed in the same volume of buffer O (20 mM Tris-HCl pH 7.5, 195 mM Sorbitol, 5 mM NaCl, 1 mM $MgCl_2$) and the centrifugation step was repeated. Finally, the proteoliposome pellet was resuspended in a minimal volume of buffer O to achieve a lipid concentration of -100 mg/ml. Activation or inactivation was further triggered by the addition of 10 mM ATP or 10 mM ADP. Furthermore, 14% deuterated glycerol (Glycerol-d8) was added to each sample. To finally initiate $K^+$ flux at activating conditions, 1 µM CCCP was added, shortly before 15 µl of the sample were transferred into an EPR quartz tube with a 1.6 mm outer diameter and flash frozen in liquid nitrogen.

### Pulsed-EPR measurements

Four-pulse DEER measurements were performed on a Bruker ELEXSYS E580 EPR spectrometer operating at Q-band frequencies (~34 GHz). The spectrometer was equipped with a PELDOR unit (E580-400U, Bruker), an ELEXSYS SuperQ-FT accessory unit, an arbitrary waveform generator (SpinJet-AWG), a Bruker AmpQ 150 W TWT amplifier, and a Bruker EN5107D2 dielectric resonator. The temperature was adjusted to 50 K by a continuous-flow helium cryostat (CF935, Oxford Instruments) and a temperature control system (ITC 502, Oxford Instruments). Measurements were acquired with a dead-

time free sequence of Gaussian pulses and a 16-step phase cycling[54,55]. The pulse lengths were 38 ns for the pump pulse and 48 ns for the observer pulses, respectively. The pump pulse was applied to the maximum of the field-swept spectrum, and the observer pulses were set at 80 MHz lower. The shot repetition time (SRT) was 2 ms. The deuterium modulations were suppressed by increasing the first inter-pulse delay by 16 ns over 8 steps. The distance distributions were obtained by subtracting the background function from the normalized primary DEER data V(t)/V(0) using the MATLAB-based software package DeerAnalysis2022[56]. The resulting form factors F(t)/F(0) were fitted to distance distributions by a model-free Tikhonov regularization approach using the L-curve criterion. To determine the uncertainty for the probability distribution, distances for different background functions were obtained by a variation of the starting time and the dimensionality for the spin distribution.

## Cryo-EM sample preparation
For cryo-EM CF-1.2/1.3-4Cu holey carbon grids (Protochips, Inc.) were glow discharged twice with a PELCO easiGlow device at 15 mA for 45 s. KtrAB sample was supplemented with 1 mM ATP and 2 mM $MgCl_2$ prior to freezing. A volume of 3 μl of sample (9 mg/ml) was applied on a grid immediately before plunge freezing. Samples were blotted at 4 °C and 100% humidity using a Vitrobot Mark IV device (Thermo Scientific) with a blot force of 4 for 12 s, before plunge freezing in liquid ethane. Vitrified samples were stored in liquid nitrogen.

## Cryo-EM image recording and processing
Images were recorded using a Titan Krios G2 microscope (Thermo Scientific) operated at 300 kV in Energy-Filtered Transmission Electron Microscopy (EFTEM) mode. Electron-optical alignments were adjusted with EPU software (Thermo Scientific). Micrographs were recorded using automated data acquisition in EPU (2.3) on a Gatan K3 direct electron detector in electron counting mode at a nominal magnification of X105,000, corresponding to a calibrated pixel size of 0.831 Å. Dose fractionated movies were recorded at an electron flux of 13 e⁻ × pixel$^{-1}$ x s$^{-1}$ for 2 s exposure time, corresponding to a total dose of ~40 e⁻/Å$^2$. A defocus range between -0.5 and -2.5 μm was used. To correct beam-induced motion and to generate dose-weighted images, MotionCor2 was used[57]. Next, CTFfind4.1 was used to determine the contrast transfer function (CTF) parameters[58]. Particles were automatically picked by crYOLO[59]. Cleaning the datasets by 2D and 3D classification, 3D map refinement, Bayesian polishing and correction for electron optical aberrations were performed using RELION-3.1[60,61]. D2 symmetry was applied in case of the $KtrB_2KtrA_8KtrB_2$ complex, and C2 symmetry for the $KtrB_2KtrA_8$ complex. The maps were further improved by density modification in Phenix[62].

## Model building
Atomic models were built in Coot[63] and refined by real-space refinement in Phenix[64]. As an initial model served the structure based on the 6.6 Å cryo-EM density map (EMD-3450) obtained by Diskowski et al. (2017)[18].

## Purification of KtrA
For the purification of KtrAC3H, the cytosolic fraction was isolated as described above. The KtrAC3H-containing supernatant was incubated with 1 ml/(l cell culture) of pre-equilibrated Ni$^{2+}$-NTA agarose in the presence of 10 mM imidazole for at least 3 h at 4 °C under gentle agitation. After 3 h of binding, unbound proteins were removed by transferring the mixture to a chromatography column. The agarose was washed with 50x CV buffer W supplemented with 50 mM imidazole. Finally, the protein was eluted once with 1/2x CV and 4 times with 1x CV of buffer W containing 500 mM imidazole. The protein sample was further purified by size exclusion chromatography using a Superdex200Increase 10/300 GL column (GE Healthcare) equilibrated

with buffer W or ITC buffer (25 mM HEPES pH 8, 140 mM NaCl, 60 mM KCl), depending on the purpose.

## Isothermal titration calorimetry (ITC)
Purified KtrA was concentrated to 40–85 μM in ITC buffer (20 mM HEPES·HCl pH 8, 140 mM NaCl, 60 mM KCl). ITC measurements were performed at 24 °C with a MicroCal iTC200 System (Malvern) to determine affinities towards ATP or ADP. The cell was loaded with 280 μl protein solution and after an initial titration step of 0.2 μl, the ligand was further titrated with a volume of 2 μl. In total 20 injections were performed with 3 min intervals between each injection. The syringe speed was set to 750 rpm. For analysis MicroCal ITC-ORIGIN Analysis Software was used and binding parameters were determined by a single-site binding fit. Exact concentrations are indicated in the figure description.

## Differential scanning fluorimetry (DSF)
DSF measurements were performed to determine the effect of nucleotide binding on the thermostability of purified KtrA. To follow protein unfolding, the fluorescent dye SYPRO™ Orange was used, which is sensitive to exposed hydrophobic regions of the protein. Samples were prepared containing 5 μM purified KtrA (monomer) in buffer W. Prior to the measurements, the samples were supplemented with different concentrations of ADP and ATP. The total sample volume was adjusted to 27 μl, to which 3 μl of 100x SYPRO™ Orange dye were added. Measurements were executed using the Rotor-Gene Q 5plex HRM System (Qiagen). The SYPRO™ Orange dye was excited at a wavelength of 470 nm, and emission was measured at a wavelength of 555 nm. Before heating, the samples were kept at the pre-melting temperature of 25 °C for 90 s. Subsequently, the temperature was increased by 1 °C every 60 s, and the fluorescence intensity was recorded at temperatures between 25 °C and 80 °C. The gain was set to 5. For the competition assay, the protein was either preincubated with 500 μM ADP or ATP. Subsequently, the competitor was added at different concentrations.

## Negative staining and 2D classification
To determine conformational changes in KtrA, 3 μl of the purified sample was adjusted to a concentration of 0.006 mg/ml supplemented with either 1 mM ATP + 2 mM $MgCl_2$ or 1 mM ADP + 2 mM $MgCl_2$. Subsequently, the samples were applied to glow-discharged carbon-coated grids (20240 C, Cu, 400 mesh, SPI). Glow discharging was performed with a PELCO easiGlow device (Ted Pella, Inc.) at 15 mA for 45 s. The applied sample was incubated on the grid for 30 s at room temperature. Subsequently, the sample was removed with filter paper Nr. 4 (Whatman®) and 3 μl of 2% uranyl formate were applied and immediately removed with filter paper. Final staining was achieved by applying 3 μl of uranyl formate and incubating for 1 min. The stain was removed with filter paper and the grid was air-dried. Grids were investigated in a Tecnai G2 Spirit BioTWIN microscope (FEI Company) with a Rio™ 16 camera (Gatan, Inc.). Micrographs were collected by automated acquisition using Leginon[65]. Automated particle picking and 2D classification were performed with cisTEM[66]. The 2D class averages obtained from negative staining were used to analyse the effect ATP and ADP on the conformation of KtrA. To determine if KtrA showed the oval or the square conformation, the longest and shortest diameter of each class was measured using GIMP (gimp.org). The ratio of the diameters was calculated and compared to a set threshold of 1.115. Classes with a diameter ratio of less than 1.115 were defined to show the square conformation. For classes with a ratio of more than 1.115, the oval conformation was assumed.

## Whole-cell K$^+$ uptake experiments
Whole-cell uptake experiments were performed according to Tholema et al. (2005)[67]. In detail, *E. coli* LB2003 cells transformed with the

expression plasmids encoding for KtrAB (pKT84[15]) or variants thereof and grown in K3 minimal medium supplemented with 100 µg/ml ampicillin and 0.02% arabinose at 37 °C o/d at 180 rpm shaking. At an OD$_{600}$ of 0.6–0.8 the cells were harvested at 5,500 g for 10 min and cell pellet was washed twice in 10 ml 120 mM Tris-HCl pH 8. Cells were adjusted to an OD$_{600}$ of 30 and incubated for 5 min at 37 °C in a water bath under constant shaking. To deplete cells from internal potassium and sodium, 1 mM EDTA was added, and cells were incubated at 37 °C for exactly 7 min shaking at 130 rpm in a water bath. To completely remove the ions and the residual EDTA, the cells were washed 3x with 200 mM HEPES-triethanolamine (TEA) pH 7.5. Finally, the OD$_{600}$ was adjusted to 30 in 200 mM HEPES-TEA pH 7.5.

Uptake experiments were performed in a water bath at 25 °C under constant shaking. For the experiment, cells were diluted 1:10 in 10 ml HEPES-TEA pH 7.5. Initially, the cells were energized with 10 mM glucose and incubated for 10 min. K$^+$ uptake was initiated by the addition of different KCl concentrations (0.1, 0.2, 0.4 and 1 mM). All measurements were performed in the presence of 5 mM NaCl. Prior to KCl addition, 2×1 ml samples were taken to determine K$^+$ contamination and within 10 min after KCl addition, 1 ml samples were taken at different timepoints (1 min, 2 min, 3 min, 4 min, 7 min, 10 min). K$^+$ uptake was stopped by centrifugation of the sample through 200 µl of silicone oil (1.04 g/cm$^3$) in a 1.5 ml reaction tube for 2 min at 4100 g. This separated the cells from buffer. Buffer and silicone oil were removed with a vacuum pump and the tip of the reaction tube, which contained the cell pellet, was cut off and transferred in 1 ml 5% TCA. After the pellet was dissolved, cells were disrupted by freezing at -20 °C overnight, followed by cooking at 90 °C for 10 min. Subsequently, 3 ml of 6.7 mM CsCl$_2$ were added and the K$^+$ concentration was determined by flame photometry (ELEX 6361, Eppendorf). The Michaelis-Menten constants (K$_M$) and the maximum uptake velocities (V$_{max}$) were calculated by using the Michaelis-Menten fit in GraphPad Prism5 where the following equation was applied: Y = V$_{max}$*X/(K$_M$ + X).

### Molecular dynamics simulations

The KtrB and KtrAB structures (PDB ID 7zp9) were placed in a heterogenous bilayer comprising POPE (45%), POPG (40%), and cardiolipin (POCL) (15%) using the CHARMM-GUI webserver[68]. Sodium and chloride ions were added to establish an ion concentration of 150 mM. The all-atom CHARMM36m force field was used for proteins, lipids, and ions, together with the TIP3P model for water molecules[69–73] MD trajectories were analysed using Visual Molecular Dynamics (VMD)[74] and MDAnalysis[75].

We performed all simulations using GROMACS 2021.2[76]. The initial setups were energy-minimized for 5000 steepest descent steps and equilibrated for 1.5 ns in a canonical (NVT) ensemble and afterwards for 7 ns in an isothermal-isobaric (NPT) ensemble under periodic boundary conditions. The restraints on the positions of nonhydrogen protein atoms of initially 4000 kJ·mol$^{-1}$·nm$^2$ were gradually released during equilibration. The cutoff distance for non-bonded interactions (van der Waals and Coulomb interactions) was set to 1.2 nm. Particle-mesh Ewald summation[77] was used to treat long-range electrostatic interactions with cubic interpolation and a 0.12-nm grid spacing. The time step was initially 1 fs in the NVT equilibration and was increased to 2 fs during the NPT equilibration. The LINCS algorithm was used to fix all bond lengths[78]. During the equilibration phase, constant temperature and pressure were established with a Berendsen thermostat, combined with a coupling constant of 1.0 ps and a semi-isotropic Berendsen barostat with a compressibility of 4.5 × 10$^{-5}$ bar$^{-1}$, respectively[79]. The Berendsen thermostat and barostat were replaced by a Nosé−Hoover thermostat and a Parrinello−Rahman barostat during the production runs[80,81]. Several unrestrained replicate simulations of KtrB and KtrAB were performed at temperatures of 310, 353, and 395 K, as listed in see Suppl. Table 2.

Native contacts are defined[82,83] as the contacts present within the initial structure. Two heavy atoms i (from N-termini residue 7-13) and j (from KtrA) are considered to form a native contact if their distance r$_{ij}^0$ in the initial structure is less than 4.5 Å. The fraction of native contacts Q is then defined in a configuration X as

$$Q\left(r_{ij}(X), r_{ij}^0\right) = \frac{1}{N} \sum_{(i,j)} \frac{1}{1 + e^{(\beta(r_{ij}(X) - \lambda r_{ij}^0))}} \quad (1)$$

where the sum runs over the N distinct pairs of native contacts (i,j) and r$_{ij}$(X) is the distance between heavy atoms i and j in configuration X. We set the smoothing and padding parameters to β = 5 Å$^{-1}$ and λ = 1.5, respectively.

To investigate the interaction of N-termini with the membrane, we calculated the number of contacts between N-termini amino acids and lipid molecules. An N-termini residue and a lipid molecule were considered to be in contact when at least one heavy-atom pair was within 3.5 Å in distance[82].

### Multiple sequence alignment

Protein sequences of several KtrBs of each Gram-positive and Gram-negative species were obtained from UniProt. Using PROMAS3D server, the length of the N-terminus and the start of the first transmembrane helix was predicted[84]. The multiple sequence alignment was performed using the Clustal Omega server[85]. JalView 2.11.2.7 was used for further analysis and visualization[86].

### Reporting summary

Further information on research design is available in the Nature Portfolio Reporting Summary linked to this article.

### Data availability

The new cryo-EM maps and models described in this article have been deposited in the wwPDB with accession codes EMD-14851 and 7zp9 for the KtrB$_2$A$_8$B$_2$ complex, EMD-14859 and 7zpo for the native KtrB$_2$A$_8$ complex, and EMD-14862 and 7zpr for the KtrB$_2$A$_8$B$_2$ complex with N-terminal KtrB deletion. Additionally, published PDB 4j91 has been used for structural comparison. Molecular dynamics simulations performed in this study have been deposited in Zenodo [https://doi.org/10.5281/zenodo.14501218]. Source data are provided with this paper.

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

## Acknowledgements

We thank the Central Electron Microscopy Facility of the Max Planck Institute of Biophysics for cryo-EM infrastructure and technical support, and Juan Castillo-Hernández and Özkan Yildiz for support in cryo-EM data processing. We thank Katja Ohlemüller for performing whole cell uptake assays. This work was supported by the German Research Foundation via SPP1879 to JV and IH (VO 1449/1-1 and HA 6322/4-1), Emmy Noether grant to IH (HA6322/3-1), CRC807 (P22: IH, P25: GH), CRC1507 "Membrane-associated Protein Assemblies, Machineries, and Supercomplexes" – Project ID 450648163 (P4: IH, P12: GH, P14: JV), and the Heisenberg programme to IH (HA 6322/5-1). G.H. and J.V. thank the Max Planck Society for support.

## Author contributions

J.S. and I.H. conceived this study. All authors designed the experiments. J.S. and D.G. performed in vivo experiments. J.S., D.G., C.T., and M.S. purified KtrA and KtrAB for EM and EPR analysis. J.S., S.Ka., D.J.M., and J.V. performed the EM analyses, and built and validated the atomic models. D.W. and S.Ke. performed EPR analysis. A.R.M. performed MD simulations. All authors participated in the data analysis. J.S. wrote the initial draft, all authors participated in manuscript editing and revision. G.H., J.V., and I.H. supervised work and secured the funding for this work.

## Funding

## Competing interests

The authors declare no competing interests.
