## [Transparent Peer Review file · Nature Communications]

A short intrinsically disordered region at KtrB's N-terminus facilitates allosteric regulation of K⁺ channel KtrAB

Corresponding Author: Professor Inga Hänel

Version 0:

Reviewer comments:

Reviewer #1

(Remarks to the Author)

The authors have put together an interesting manuscript to elucidate the role of the KtrB N-termini in channel opening. I commend the use of a variety of approaches to address the issue and note that the cryo-EM maps are of very good quality and generally supportive of the authors' claims. However, I have some resounding concerns and comments on the current version of the manuscript that I have outlined below:

MAJOR:

-In Fig.4d, Supplementary Fig.7, and Supplementary Fig. 9, no error bars are shown for the concentration series, but the authors note the assay was run in triplicate, and the resulting Michaelis-Menton plot has error bars. Show all data, either composite with error bars, or individual traces for the uptake experiment. Or alternatively, note that the uptake graph is only representative.

-In Fig. 5b it seems there is a mismatch between the model and the cryoEM density shown. The overall model/view shown is that of 5a, but the density of KtrB shown is from the view/model shown in 5c. I believe the authors meant to show the overall model in 5c. Please fix, or if intentional, please make it clear in the legend which models are shown and the intent for the overlay in 5b.

-If making a comparison between 5a and 5c, the same viewing angle should be shown, with a note in the legend of how the structures were aligned.

-Fig. 5b the cryoEM volume should be made transparent to reveal the modeling of residues.

-The authors treat their MD analysis of the 2 subunits as if they were separate MD runs. It seems this ignores the possibility that there are possible allosteric effects between the two subunit results. I would defer to the MD expert reviewer, but in my opinion, 1 simulation run of only 1 micro sec is insufficient in replication and in time for supporting conclusions.

-There is a discrepancy between the results of the DSF assay in which the authors observed successful displacement of ADP or ATP for the other nucleotide and the speculation that DEER measurements resulted from ADP being unable to be released by ATP. Could the authors elaborate on why the discrepancy may exist or why they have come to their conclusion despite the discrepancy? (Is it due to limitations or differences in the techniques?)

-Could the authors conduct pulsed EPR experiments on the detergent-solubilized complex, as well as, the liposome-reconstituted complex? It would help substantiate the author's claims regarding the role of the membrane.

-In the data availability statement, the EMDB accession code associated with PDB 7ZP9 is wrong, 'EMD-14841' is listed, but it should be 'EMD-14851'

MINOR:

-Authors should add labels for the subunits in Fig. 1 and 2.

-The words "Fehler! Linkreferenz ungültig." appear on line 72, presumably due to an error in referencing.

-In Fig 2. add degrees of rotation to the arrow

-Authors note on line 194, "Unfortunately, a high ATP concentration results in background artifacts in cryoEM, hampering data analysis". Could the authors cite their source or expand on the conditions attempted? For example, 'high' is a relative term; please add a value (e.g., "ATP concentrations above XXXX mM result in background artifacts...")

-Have the authors tested whether exogenous peptide of the N-termini can activate the channel? I'm also curious if they made the opposite charge mutations to Aspartate or Glutamate in K16/R17/K19/K21 in addition to the neutral Alanine mutations shown.

-Associated PDB codes and EMD codes should be added to Sup. Table 1

-In the models, there are MANY residues with side chains modeled outside the resolvable EM density. For some, the residues are entirely in the wrong position/rotamer while most should simply be stubbed at C β . I have listed this in the minor comments as I note that the areas around where the authors have made conclusions are modeled well enough so as not to affect the conclusions of the paper, but I caution the authors to be more conservative in their inclusion of side chains and careful in checking the overall structures to improve the trusted quality of their deposited models.

Reviewer #2

(Remarks to the Author)

In this work, Stautz and coworkers report a Cryo-EM structure of the inactive state of the KtrAB complex. By integrating their study with further Cryo-EM structure of KtrAB with N-termini deleted, functional characterization of several mutants, and Molecular Dynamics (MD) simulations, the authors provide a detailed picture of the gating mechanism of the KtrAB complex.

While the results of the experimental part are robust and convincing, overall, I found the computational part slightly disappointing. It is true that the MD simulations are only used to confirm evidence gathered by multidisciplinary experiments, but I cannot help but notice that a little additional effort would have been enough to make computational results more robust.

For example, in Figure 7b, configurations of the N-termini taken at regular time intervals are shown to support the fact that this region is disordered without the presence of the KtrA ring, but it would have been desirable to see a more detailed characterization of the conformational space, especially compared to that explored by the same portion in the complex. The number of contacts between this region and the lipids as well as the native contacts within the complex are not particularly enlightening in this regard.

More generally, it is disappointing that only one simulation was carried out for the two systems, while the importance of replicates is widely recognized even in the context of molecular simulation experiments.

Furthermore, I would refrain from claiming that what is proposed is an "allosteric network", unless a network is actually used to characterize the allosteric effect, possibly by analyzing MD trajectories. Without such kind of characterization, perhaps it would be more appropriate to simply refer to it as an "allosteric mechanism".

As a final note, I found the Methods section concerning MD simulations lacking information regarding minor, but still important, parameters for data reproducibility, like the cutoffs used or the parameters relevant for controlling the temperature and pressure of simulations.

Reviewer #3

(Remarks to the Author)

Stautz et al using a combination of cryoEM, EPR and MD highlight the role of a disordered loop in the gating of the KtrAB channel, which are important in K⁺ homeostasis in bacteria. These channels contain cytosolic RCK-domains and binding of nucleotides within these domains is key to their regulation. This particular flexible loop could not previously resolved and here is only partially resolved by cryoEM. Notably, experimental EPR (DEER/PELDOR) distance measurements come as evidence in support of this loop's role in the complex gating of KtrAB channels. EPR offers here wider condition flexibility including measurements performed in the presence of lipid bilayers. KtrAB is a challenging system to study and authors used an appropriate method toolkit to address it. Since, key findings rely on the EPR data, additional analysis and modelling is required, including re-measurement(s) and/or longer averaging to obtain higher SNR data. This is essential to conclude the relative proportion of the ensemble between the two KtrAB states (ATP vs ADP). It is encouraging that EPR can pick up such differences (especially when these are only few Angstroms). Authors can still report on the change, as EPR detects shifts in the equilibrium, but more Cys-pairs and/or better SNR is needed to resolve the background, before any semi-quantitation claim for the KtrAB conformational ensemble could be made.

Major points:

1) "Both in the presence of 10 mM ATP and 10 mM ADP a weak oscillation in the dipolar evolution function and consequently a broad distance distribution was observed, indicating high flexibility of the intramembrane loop (Fig. 3)"

Based on the data presented I would say there is no oscillation. Further, the data are undersmoothed (L-curve should be also presented in this case) and these multiple peaks are not "broad distributions". If authors want to claim the existence of a broad distribution then they need to provide a more rigorous and detailed EPR data analysis (Deerlab, Deernet and DeerAnalysis validation including fitting parameters).

2) "In detail, in the presence of ADP, a distance distribution ranging from 1.5 to 4 nm with a major peak at 3.3 nm was revealed. This peak corresponds to the intramembrane loop in the closed conformation,"

As per point 1, according to these EPR data this cannot be claimed. Depending on the fitting parameters and selected background multiple peaks at different positions between 20-40Å could be generated.

3) "This peak corresponds to the intramembrane loop in the closed conformation,"

This is only an indication of a dynamic-flexible loop rather than a specific state. Since no (or extremely weak) oscillation is there, data should be treated with extreme caution. Authors can still report on the EPR changes they observe when nucleotides are present by overlaying raw data on same graphs. Also, *in silico* modelling should be presented. I could count three species/states in the distribution. Why is the 3.3 nm the right one and all the rest should be ignored?

4) "Although all KtrAB preparations were done in the presence of ATP, the cryo-EM maps clearly display features that define the ADP-bound state,"

Previous literature suggests that AMP co-purifies in other RCK-domain full length proteins, suggesting it is important to protein's structural integrity, while in other studies ATP was added to enable crystallisation and saturate the RCK-domain sites, but although it was modelled in as ATP, inspection of the actual density resembles more of AMP/ADP presence ((2006) Cell 126: 1147-1159; (2017) Biochemistry 56: 4219-4234)). Do these observations have any implication on the occupancy of the KtrAB RCK-domain sites by ATP/ADP?

5) "it appears that neither conformational changes in the KtrA ring nor the exchange of nucleotides in all binding sites occurs in the KtrAB complex"

Again linked to previous point 4, what is the experimental evidence that this exchange has taken place? If it does, is it possible to detect a conformational change by EPR?

6) T318C spin labelled position is not currently shown. I understand this site is on a disordered loop but still a model should be used to see how this distance is possible. If only 4 Å differences are reported here, this may just be a small change in label orientation. This study relies on the EPR data as cryoEM cannot definitively conclude this (i.e. resolves part of it). I appreciate the EPR data presented here and they indeed support the authors' argument that the loop becomes more rigid/ordered/defined in the presence of the bilayer. The EPR/DEER data suggest, i.e. a broad distribution (dynamic/disordered) becomes narrow (oscillating trace resulting in narrow distribution). Thus, I believe the strength of the current EPR data is on reporting this particular change, rather than assigning specific conformational states.

7) "To incorporate ADP or ATP into the liposomes, 10 mM ADP and 10 mM ATP, respectively, were added to the samples and three cycles of freeze and thaw were performed"

How do the authors control (or test) for channel reconstitution orientation as this for KtrAB will affect atp/adp binding through RCK domain accessibility

8) Previous studies relied on Pulse EPR to successfully assign conformation in other multimeric channels but not much is mentioned here, given the field has grown substantially over the last years.

9) "In the presence of ADP, a clear oscillation in the experimental raw data and the background corrected dipolar evolution function($F(t)$) was visible, resulting in a narrow distance distribution centred at 3.3 nm with only a small second fraction centred at 3.7 nm."

Data are quite noisy to claim this and small changes in the background (also second visible, but very noisy oscillation ~1.2 us) would alter the relative proportion of the two states/species observed. Authors should either average longer with same time windows to obtain less noisy data or repeat with higher protein concentration.

10) "In the presence of ATP no clear shift in the mean distance distribution towards the open state occurred, similar to the detergent solubilised sample"

Again, as per previous point, this is difficult (or currently not possible) to tell. I agree there is a difference between ATP and

ADP traces and that resulting distances are shifted towards longer distances (i.e. ATP) and this is remarkable it has been picked by EPR. However, since this is still an ensemble of multiple states in order to conclude the relative proportion between these two states requires rigorous data analysis and direct comparison between the raw data with similar SNR. Relative proportion of states with current analysis and data cannot be claimed in a quantitative way, given that “distance peak intensities” are currently prone to large errors and depend on the analysis performed.

Minor

- 1) Fig S8: asterisk band although mentioned no gel presented
- 2) Line 72: there is typo. please remove
- 3) DEER method section requires more detail on how these experiments were set up and data were processed and analysed. For instance the errors in the presented distributions it is not described how these were derived.
- 4) 2 us time windows are used in the EPR measurements and although these are relatively short, are still is appropriate for the distances presented (i.e. complete oscillation is seen in the raw data), despite some of them being noisy. However, a traffic light bar (DeerAnalysis) should be included to appreciate confidence level for the presented distance distributions.

Version 1:

Reviewer comments:

Reviewer #1

(Remarks to the Author)

The revised version of the manuscript by Stautz et. al. on the mechanism of ATP-mediated KtrAB channel opening is improved from the initial version. The additional experiments, including the expansion of molecular dynamic studies, and the re-writing of the manuscript have made for a clearer picture of the mechanistic details. Further transparency and clarity of quality control benchmarks also improved in this revision. In short, the authors have adequately and thoroughly revised the manuscript and updated related PDB models in response to my previous comments. I have no additional concerns.

Reviewer #2

(Remarks to the Author)

The authors have thoroughly addressed all of my concerns.

Reviewer #3

(Remarks to the Author)

I am happy with the changes introduced in the revised manuscript. Now the EPR data are consistent with the proposed KtrAB gating model.

Dear editor, dear reviewers,

we thank the reviewers for their thorough assessment of our work and their comments, suggestions, and critiques. In response, we have critically re-evaluated our data and revised the manuscript accordingly, which in our opinion has significantly improved as a result. We re-organized the manuscript, removed less relevant data (formerly Figure 3) and instead added more relevant measurements (Figures 7: further MD simulation and 8b: EPR with KtrB alone). We also further highlighted data that further support our model of KtrABs regulation. In the following, we respond in detail to all issues raised.

REVIEWER COMMENTS

Reviewer #1 (Remarks to the Author):

The authors have put together an interesting manuscript to elucidate the role of the KtrB N-termini in channel opening. I commend the use of a variety of approaches to address the issue and note that the cryo-EM maps are of very good quality and generally supportive of the authors' claims. However, I have some resounding concerns and comments on the current version of the manuscript that I have outlined below:

MAJOR:

-In Fig.4d, Supplementary Fig.7, and Supplementary Fig. 9, no error bars are shown for the concentration series, but the authors note the assay was run in triplicate, and the resulting Michaelis-Menton plot has error bars. Show all data, either composite with error bars, or individual traces for the uptake experiment. Or alternatively, note that the uptake graph is only representative.

We now clarify that only representative uptake graphs are shown in Fig. 4d, Supplementary Fig.7 (now 6) and Supplementary Fig.9 (now 10). The caption of Figure 4 now states; 'Uptake experiments were performed in biological triplicates with one representative graph shown.' For the Supplementary Fig. 6 and Fig. 10 the description states: 'Each set of experiments was performed in biological triplicates, with representative results from a single set shown. ... Error bars represent SDs between the triplicates.' The complete set of raw data from the triplicates is included in the source data file.

-In Fig. 5b it seems there is a mismatch between the model and the cryoEM density shown. The overall model/view shown is that of 5a, but the density of KtrB shown is from the view/model shown in 5c. I believe the authors meant to show the overall model in 5c. Please fix, or if intentional, please make it clear in the legend which models are shown and the intent for the overlay in 5b.

-If making a comparison between 5a and 5c, the same viewing angle should be shown, with a note in the legend of how the structures were aligned.

We thank the reviewer for pointing out the mismatch in the representation of the models. We updated the representation with all models having the same viewing angle in Figure 5.

-Fig. 5b the cryoEM volume should be made transparent to reveal the modeling of residues.

We appreciate the reviewer's suggestion. The representation of the volume has been updated as a mesh.

-The authors treat their MD analysis of the 2 subunits as if they were separate MD runs. It seems this ignores the possibility that there are possible allosteric effects between the two subunit results. I would defer to the MD expert reviewer, but in my opinion, 1 simulation run of only 1 micro sec is insufficient in replication and in time for supporting conclusions.

We agree that one simulation was not enough to support our conclusions. Therefore, we performed additional simulations. We agree on the importance of running several replicates for simulations. To address this, we have conducted multiple replicates at three different temperatures for KtrB and KtrAB. The conditions are summarized in Supplementary Table 2. This point is further addressed in response to Reviewer #2.

-There is a discrepancy between the results of the DSF assay in which the authors observed successful displacement of ADP or ATP for the other nucleotide and the speculation that DEER measurements resulted from ADP being unable to be released by ATP. Could the authors elaborate on why the discrepancy may exist or why they have come to their conclusion despite the discrepancy? (Is it due to limitations or differences in the techniques?)

To clarify, the displacement of ADP or ATP observed in the DSF assays was performed on the KtrA subunit alone. DEER measurements on the solubilized KtrAB complex (former Fig. 3) indicate that nucleotide exchange does not occur because no changes in distance distribution were observed in the presence of ADP or ATP, suggesting a modified mechanism for nucleotide exchange in the presence of KtrB. However, we have also toned down our interpretation on the DSF displacement assay as the data clearly show that ATP much less efficiently triggers a shift from the ADP- to the ATP-bound state than vice versa. Nonetheless, when adding ATP in excess to KtrA alone, the square-shaped, ATP-bound conformation could be stabilized efficiently as shown by DSF and negative stain EM.

-Could the authors conduct pulsed EPR experiments on the detergent-solubilized complex, as well as, the liposome-reconstituted complex? It would help substantiate the author's claims regarding the role of the membrane.

We thank the reviewer for this suggestion. A comparison of the EPR measurements on the detergent-solubilized sample (previously Figure 3) and on lipid-reconstituted sample (Figure 8) pointed to a possible regulatory role of the lipid surrounding in complex activation. However, in the revised manuscript, driven by reviewer comments, we more strongly focus on nucleotide binding of KtrA as well as the functional data. From the reviewers' comments, it became clear that the EPR data for detergent-solubilized protein were rather distractive than supportive. Therefore, we added DEER measurements of the KtrB dimer alone (Fig. 9) in liposomes which, together with the more extensive MD simulations (Fig. 7) strongly support our claim of the role of KtrB's N terminus and its interaction with the membrane in the activation mechanism of KtrAB.

-In the data availability statement, the EMDB accession code associated with PDB 7ZP9 is wrong, 'EMD-14841' is listed, but it should be 'EMD-14851'

The accession codes have been updated.

MINOR:

-Authors should add labels for the subunits in Fig. 1 and 2.

We appreciate the suggestion, the labels have been added in Fig. 1 and 2.

-The words "Fehler! Linkreferenz ungultig." appear on line 72, presumably due to an error in referencing.

The missing reference Mikušević et al 2019 has been added.

-In Fig 2. add degrees of rotation to the arrow

We have updated Figures 1 and 2 as well as in Supplementary Figure 9 as suggested.

-Authors note on line 194, “Unfortunately, a high ATP concentration results in background artifacts in cryoEM, hampering data analysis”. Could the authors cite their source or expand on the conditions attempted? For example, ‘high’ is a relative term; please add a value (e.g., “ATP concentrations above XXXX mM result in background artifacts...”)

We decided to delete this sentence because the additional information distracts from what we actually want to bring across and now instead just mention that physiological concentrations of up to 5 mM ATP were used (p. 6, line 170).

-Have the authors tested whether exogenous peptide of the N-termini can activate the channel? I’m also curious if they made the opposite charge mutations to Aspartate or Glutamate in K16/R17/K19/K21 in addition to the neutral Alanine mutations shown.

We thank the reviewer for the interesting suggestions. In our current model, the N-termini are required for the pulling motion of the extended helices, which in turn leads to the activation of the complex. According to this model, the presence of the N-termini alone is not sufficient for the activation of the complex. The effect of the suggested mutations on the N-termini is very intriguing; however, it is beyond the scope of this study. We plan to introduce these mutations in our next study.

-Associated PDB codes and EMDB codes should be added to Sup. Table 1

The accession codes have been added.

-In the models, there are MANY residues with side chains modeled outside the resolvable EM density. For some, the residues are entirely in the wrong position/rotamer while most should simply be stubbed at C β . I have listed this in the minor comments as I note that the areas around where the authors have made conclusions are modeled well enough so as not to affect the conclusions of the paper, but I caution the authors to be more conservative in their inclusion of side chains and careful in checking the overall structures to improve the trusted quality of their deposited models.

The reviewer is correct; we had not removed any side chains from the model after refinement. The large majority of side chains lacking density are located on the outside of the protein and don’t affect the interpretation of the structures. However, we have now removed the side chains and uploaded the new models to the PDB.

Reviewer #2 (Remarks to the Author):

In this work, Stautz and coworkers report a Cryo-EM structure of the inactive state of the KtrAB complex. By integrating their study with further Cryo-EM structure of KtrAB with N-termini deleted, functional characterization of several mutants, and Molecular Dynamics (MD) simulations, the authors provide a detailed picture of the gating mechanism of the KtrAB complex.

While the results of the experimental part are robust and convincing, overall, I found the computational part slightly disappointing. It is true that the MD simulations are only used to confirm evidence gathered by multidisciplinary experiments, but I cannot help but notice that a little additional effort would have been enough to make computational results more robust.

For example, in Figure 7b, configurations of the N-termini taken at regular time intervals are shown to support the fact that this region is disordered without the presence of the KtrA ring, but it would have been desirable to see a more detailed characterization of the conformational space, especially compared to that explored by the same portion in the complex. The number of contacts between this region and the lipids as well as the native contacts within the complex are not particularly enlightening in this regard.

We thank the reviewer for the insightful comments and suggestions. For the revised version of the manuscript, we performed extensive additional simulations and conducted detailed computational measurements comparing KtrB with KtrAB, focusing on the total N-termini length, the distance of the N-termini from the membrane, and the number of interactions with lipids (Fig. 7g-j). These measurements were performed multiple times at three different temperatures (Supplementary Table 2). Importantly, the expanded analysis supports our model, indicating that in KtrB, the N-termini interact with the membrane in the absence of KtrA, whereas in the complex, the interaction of the N-termini shifts from the membrane to KtrA. The simulations also confirmed the role of the hydrophobic patch and positively charged amino acids as interaction anchors with lipids.

Furthermore, our analysis of the N-terminus length as measure of conformational change of this region shows that the N-termini keep the same conformation in the KtrAB simulations, while in KtrB the N-termini explore a broader distribution during the simulations (Fig. 7h). Similar conclusions can be drawn by the distance of the N-termini from the membrane, which remains the same at all temperatures, with a smaller number of interactions in the KtrAB system compared to the KtrB dimer (Fig. 7i-j). Together, these two simulations show that the intrinsically disordered N-termini can form favourable but competitive interactions with the lipid bilayer and with KtrA.

More generally, it is disappointing that only one simulation was carried out for the two systems, while the importance of replicates is widely recognized even in the context of molecular simulation experiments.

We thank the reviewer for pointing out this issue. We agree on the importance of running several replicates for simulations. To address this, we have conducted multiple replicates at three different temperatures for KtrB and KtrAB. The conditions are summarized in Supplementary Table 2.

Furthermore, I would refrain from claiming that what is proposed is an “allosteric network”, unless a network is actually used to characterize the allosteric effect, possibly by analyzing MD trajectories. Without such kind of characterization, perhaps it would be more appropriate to simply refer to it as an “allosteric mechanism”.

We appreciate the comment and updated the manuscript states: ‘We speculated that this transition might be facilitated by partial binding of ATP to KtrA and that consequently an

allosteric mechanism of nucleotide-triggered conformational changes and relocation of the N-termini is required for achieving the active, open state.' (p. 13 lines 355ff). We also substituted "allosteric mechanism" for "allosteric network" in the abstract (line 35/36) and the introduction (p. 4, line 110) and in the legend of fig. 10.

As a final note, I found the Methods section concerning MD simulations lacking information regarding minor, but still important, parameters for data reproducibility, like the cutoffs used or the parameters relevant for controlling the temperature and pressure of simulations.

The Methods section has been revised, with requested information added. A Supplementary Table 2 summarizes the conditions and numbers of performed replicates.

Reviewer #3 (Remarks to the Author):

Stautz et al using a combination of cryoEM, EPR and MD highlight the role of a disordered loop in the gating of the KtrAB channel, which are important in K⁺ homeostasis in bacteria. These channels contain cytosolic RCK-domains and binding of nucleotides within these domains is key to their regulation. This particular flexible loop could not previously resolved and here is only partially resolved by cryoEM. Notably, experimental EPR (DEER/PELDOR) distance measurements come as evidence in support of this loop's role in the complex gating of KtrAB channels. EPR offers here wider condition flexibility including measurements performed in the presence of lipid bilayers. KtrAB is a challenging system to study and authors used an appropriate method toolkit to address it. Since, key findings rely on the EPR data, additional analysis and modelling is required, including re-measurement(s) and/or longer averaging to obtain higher SNR data. This is essential to conclude the relative proportion of the ensemble between the two KtrAB states (ATP vs ADP). It is encouraging that EPR can pick up such differences (especially when these are only few Angstroms). Authors can still report on the change, as EPR detects shifts in the equilibrium, but more Cys-pairs and/or better SNR is needed to resolve the background, before any semi-quantitation claim for the KtrAB conformational ensemble could be made.

Major points:

1) "Both in the presence of 10 mM ATP and 10 mM ADP a weak oscillation in the dipolar evolution function and consequently a broad distance distribution was observed, indicating high flexibility of the intramembrane loop (Fig. 3)"

Based on the data presented I would say there is no oscillation. Further, the data are undersmoothed (L-curve should be also presented in this case) and these multiple peaks are not "broad distributions". If authors want to claim the existence of a broad distribution then they need to provide a more rigorous and detailed EPR data analysis (Deerlab, Deernet and DeerAnalysis validation including fitting parameters).

We thank the reviewer for the insights, and we understand the criticism. Therefore, and due to restructuring the manuscript putting less emphasis on the EPR data in general, we removed the described data (previously Figure 3) and, to explore the loop flexibility, focused our manuscript more on the extended MD simulations, the whole cell uptake assays as well as the DSF measurements.

2) "In detail, in the presence of ADP, a distance distribution ranging from 1.5 to 4 nm with a major peak at 3.3 nm was revealed. This peak corresponds to the intramembrane loop in the closed conformation,"

As per point 1, according to these EPR data this cannot be claimed. Depending on the fitting parameters and selected background multiple peaks at different positions between 20-40Å could be generated.

As mentioned above, the data on detergent-solubilized KtrAB have been entirely removed from the manuscript because of the reviewer's feedback.

3) "This peak corresponds to the intramembrane loop in the closed conformation,"

This is only an indication of a dynamic-flexible loop rather than a specific state. Since no (or extremely weak) oscillation is there, data should be treated with extreme caution. Authors can still report on the EPR changes they observe when nucleotides are present by overlaying raw data on same graphs. Also, in silico modelling should be presented. I could count three species/states in the distribution. Why is the 3.3 nm the right one and all the rest should be ignored?

Agreed, data removed. Nonetheless, we now included an in-silico modelling based on our ADP-bound structure to obtain a prediction for the expected distance distribution in the ADP-bound, closed state. The data are included in Suppl. Fig 9a.

4) “Although all KtrAB preparations were done in the presence of ATP, the cryo-EM maps clearly display features that define the ADP-bound state,”

Previous literature suggests that AMP co-purifies in other RCK-domain full length proteins, suggesting it is important to protein’s structural integrity, while in other studies ATP was added to enable crystallisation and saturate the RCK-domain sites, but although it was modelled in as ATP, inspection of the actual density resembles more of AMP/ADP presence ((2006) Cell 126: 1147-1159; (2017) Biochemistry 56: 4219-4234)). Do these observations have any implication on the occupancy of the KtrAB RCK-domain sites by ATP/ADP?

We thank the reviewer for bringing this to our attention. For KtrA, we are highly confident that we co-purify nucleotides, as during purification of the protein the 260/280 ratio is very high compared to other proteins with the same purity. Therefore, we are confident that the ADP we assign in our structures is pre-bound ADP which was co-purified. The cryo-EM map also shows a bound ADP (see figure below), which has to be co-purified as only ATP was added. We are actually convinced that to some extent ATP must be bound in our structure but in such minority that the additional density of the γ -phosphate averages out over the 3-D reconstruction because it presumably is randomly bound to any of the eight binding sites in the KtrA ring. This interpretation is in line with our DSF competition assay where even for the KtrA ring alone we see only an incomplete switch from an ADP-prebound state to the ATP-bound state.

Figure: KtrA nucleotide binding site with ADP bound. a,b KtrA model with corresponding cryo-EM density from different angles.

5) “it appears that neither conformational changes in the KtrA ring nor the exchange of nucleotides in all binding sites occurs in the KtrAB complex”

Again linked to previous point 4, what is the experimental evidence that this exchange has taken place? If it does, is it possible to detect a conformational change by EPR?

To clarify, our conclusion from our data is that in (detergent) solution, an exchange of ADP to ATP and the associated conformational change of the ring does not take place on the time

scale of the experiments. Our current working model for KtrAB is that two near-simultaneous triggers are necessary for the activation of the complex: the interaction of KtrB N-termini with the membrane and the binding of ATP and Na⁺ to KtrA. This activation model is supported by the whole-cell uptake assays, as well as the DSF, ITC, negative stain EM experiments as well as MD simulations described in this manuscript (Fig. 3, 7, 9; Suppl. Fig. 5, 10). The KtrAB cryo-EM model presented here was obtained in the presence of ATP; however, since no membrane for interaction was presented it shows a non-active, ADP-bound state as no nucleotide exchange took place (Fig. 2). In our opinion, an ADP-to-ATP exchange only took place in the DEER EPR measurements, where the complex was reconstituted into liposome. This was deduced from the resulting change of distance distribution between the gating intramembrane loops (Fig. 8).

6) T318C spin labelled position is not currently shown. I understand this site is on a disordered loop but still a model should be used to see how this distance is possible. If only 4 Å differences are reported here, this may just be a small change in label orientation. This study relies on the EPR data as cryoEM cannot definitively conclude this (i.e. resolves part of it). I appreciate the EPR data presented here and they indeed support the authors' argument that the loop becomes more rigid/ordered/defined in the presence of the bilayer. The EPR/DEER data suggest, i.e. a broad distribution (dynamic/disordered) becomes narrow (oscillating trace resulting in narrow distribution). Thus, I believe the strength of the current EPR data is on reporting this particular change, rather than assigning specific conformational states.

As stated before, we now performed MMM predictions of the distance between the labelled residues. The predicted narrow distance distribution is in very good agreement with the experimentally determined distance distribution for KtrAB in liposomes in the presence of ADP, confirming our structural model (cf. Suppl. Fig. 9a and Fig. 8a). The major points we want to make with the EPR measurements are (1) that in the presence of ATP the distance distribution becomes broader when compared to ADP, which strongly implies that the gating loop became more flexible and (2) that upon deletion of KtrB's N-terminus the ADP-bound state is comparable to full-length KtrAB, while in the presence of ATP the intramembrane loop distribution is clearly altered from the ADP-bound state but also from full-length KtrAB in the presence of ATP (Fig. 8a). In fact, our now additional data for KtrB alone in liposomes suggests that the loop distribution of KtrAB with deleted N-terminus is most similar to KtrB in the absence of KtrA (Fig. 8b). This conclusion is supported by whole-cell uptake data, which show that the deletion variant of KtrAB has a similarly slow V_{max} as KtrB alone (Fig. 9).

7) "To incorporate ADP or ATP into the liposomes, 10 mM ADP and 10 mM ATP, respectively, were added to the samples and three cycles of freeze and thaw were performed"

How do the authors control (or test) for channel reconstitution orientation as this for KtrAB will affect atp/adp binding through RCK domain accessibility

The incorporation of nucleotides into liposomes is performed in a buffer supplemented with either ATP or ADP, ensuring that the desired nucleotide is present both inside and outside of the liposomes. Therefore, the final orientation of KtrAB can be disregarded.

8) Previous studies relied on Pulse EPR to successfully assign conformation in other multimeric channels but not much is mentioned here, given the field has grown substantially over the last years.

We are well aware of these studies but to our knowledge all these data are on single-pored systems. In contrast, KtrAB has two parallel pores, and the challenge is to look at conformational changes in both pores. Instead, we have highlighted that we have used the same variant before (doi: <https://doi.org/10.1074/jbc.m110.139311>) to study conformational

changes in the loops by EPR spectroscopy (actually before structural data of the KtrAB complex were available).

9) “In the presence of ADP, a clear oscillation in the experimental raw data and the background corrected dipolar evolution function(F(t)) was visible, resulting in a narrow distance distribution centred at 3.3 nm with only a small second fraction centred at 3.7 nm.”

Data are quite noisy to claim this and small changes in the background (also second visible, but very noisy oscillation ~1.2 us) would alter the relative proportion of the two states/species observed. Authors should either average longer with same time windows to obtain less noisy data or repeat with higher protein concentration.

We thank the reviewer for the comment. We performed two more sets of measurements with new samples. The SNR was improved, but in general resulted in a comparable distance distribution (Fig. 8). As mentioned above, particularly in the presence of ADP the distance distribution obtained is in very good agreement with the rotamer library analysis performed based on our ADP-bound structural model.

10) “In the presence of ATP no clear shift in the mean distance distribution towards the open state occurred, similar to the detergent solubilised sample”

Again, as per previous point, this is difficult (or currently not possible) to tell. I agree there is a difference between ATP and ADP traces and that resulting distances are shifted towards longer distances (i.e. ATP) and this is remarkable it has been picked by EPR. However, since this is still an ensemble of multiple states in order to conclude the relative proportion between these two states requires rigorous data analysis and direct comparison between the raw data with similar SNR. Relative proportion of states with current analysis and data cannot be claimed in a quantitative way, given that “distance peak intensities” are currently prone to large errors and depend on the analysis performed.

We agree with the reviewer that we were overambitious when assigning a specific distance to the ATP-bound, open state and analysing relative proportions. Hence, we toned down our interpretation and now stipulate that upon ATP addition to full-length KtrAB we get a more flexible loop. It reads: ‘In the presence of ATP, the mean distance distribution was significantly broadened towards longer distances. We conclude that the broadening reflects a more flexible gate that adopts different states to facilitate potassium ion flux. Consequently, in the liposome-based system, the addition of ATP is sufficient to trigger conformational changes’ (p. 14, lines 396ff). In our opinion, this interpretation also makes more sense because it remains unclear whether an actually stabilized open gate ever exists and in any case, we would expect an equilibrium between open and closed states. For variant KtrABΔ2-19, it in our opinion is now clear from the much-improved EPR data that the distance distribution is even broader and does not reflect the major distances present in full-length KtrAB. We attribute the featureless distribution to a dysregulated channel as also found for KtrB in the absence of KtrA (cf. p. 15, lines 401ff.).

Minor

1) Fig S8: asterisk band although mentioned no gel presented

The asterisk in this figure highlights the elution volume of the free KtrA ring; its sample is not shown in the gel.

2) Line 72: there is typo. please remove

The citation has been updated.

3) DEER method section requires more detail on how these experiments were set up and

data were processed and analysed. For instance the errors in the presented distributions it is not described how these were derived.

The Methods section has been revised.

4) 2 us time windows are used in the EPR measurements and although these are relatively short, are still is appropriate for the distances presented (i.e. complete oscillation is seen in the raw data), despite some of them being noisy. However, a traffic light bar (DeerAnalysis) should be included to appreciate confidence level for the presented distance distributions.

The 'traffic lights' were now included in the manuscript. Also, with the new samples we could apply a slightly longer time window of 2.3 μ s.